# Analytical solutions for mantle flow in cylindrical and spherical shells

Stephan C. Kramer[1], D. Rhodri Davies[2], and Cian R. Wilson[3]

[1]Department of Earth Science and Engineering, Imperial College London, UK.
[2]Research School of Earth Sciences, The Australian National University, Canberra, Australia.
[3]Earth and Planets Laboratory, Carnegie Institution for Science, Washington, DC, USA.

**Correspondence:** Stephan Kramer (s.kramer@imperial.ac.uk)

**Abstract.** Computational models of mantle convection must accurately represent curved boundaries and the associated boundary conditions of a 3-D spherical shell, bounded by Earth's surface and the core-mantle boundary. This is also true for comparable models in a simplified 2-D cylindrical geometry. It is of fundamental importance that the codes underlying these models are carefully verified prior to their application in a geodynamical context, for which comparisons against analytical solutions are an indispensable tool. However, analytical solutions for the Stokes equations in these geometries, based upon simple source terms that adhere to physically realistic boundary conditions, are often complex and difficult to derive. In this paper, we present the analytical solutions for a smooth polynomial source and a delta-function forcing, in combination with free-slip and zero-slip boundary conditions, for both 2-D cylindrical and 3-D spherical-shell domains. We study the convergence of the Taylor Hood (P2-P1) discretisation with respect to these solutions, within the finite element computational modelling framework Fluidity, and discuss an issue of suboptimal convergence in the presence of discontinuities. To facilitate the verification of numerical codes across the wider community, we provide a python package, Assess, that evaluates the analytical solutions at arbitrary points of the domain.

## 1 Introduction

Mantle convection transports Earth's internal heat to its surface: it is the 'engine' driving our dynamic Earth (e.g. Davies, 1999). The structure, composition and flow-regime within the mantle is reflected in near-surface phenomena such as plate tectonics, mountain building, dynamic topography, sea-level change, volcanism and the activity of Earth's magnetic field (e.g. Morgan, 1972; Mitrovica et al., 1989; Gurnis, 1993; Olson et al., 2013; Kloecking et al., 2019; Davies et al., 2019). The grand-challenge is to understand the operation of this giant heat engine over geologic time and its relationship to the surface geological record.

Computational modelling is one of the primary tools available for tackling this challenge. Whilst 2- and 3-D numerical models of mantle convection processes in Cartesian domains have provided important insights into a range of mantle processes (e.g. McKenzie et al., 1974; Gurnis and Davies, 1986; Davies and Stevenson, 1992; Moresi and Solomatov, 1995; van Keken et al., 2002; Hunt et al., 2012; Garel et al., 2014; Davies et al., 2016; Jones et al., 2016, 2019), 3-D spherical geometry is required to simulate global mantle dynamics. Global 3-D spherical mantle convection models, and studies focussing on their application, are now in common use (e.g. Baumgardner, 1985; Tackley et al., 1993; Bunge et al., 1996, 1997; Zhong et al.,

2000; Oldham and Davies, 2004; McNamara and Zhong, 2005; Choblet et al., 2007; Zhong et al., 2008; Tackley, 2008; Davies and Davies, 2009; Wolstencroft et al., 2009; Stadler et al., 2010; Tan et al., 2011; Kronbichler et al., 2012; Davies et al., 2013; Burstedde et al., 2013; Heister et al., 2017; Dannberg and Gassmoller, 2018; Stotz et al., 2018). However, the use of this geometry for calculations at a realistic convective vigour remains expensive. As a consequence, simplifying geometries are often used, including the axisymmetric spherical shell (e.g. Solheim and Peltier, 1994; van Keken and Yuen, 1995), the 2-D

cylinder (e.g. Jarvis, 1993; van Keken and Ballentine, 1998, 1999; van Keken, 2001; Nakagawa and Tackley, 2005) and the spherical annulus (e.g. Hernlund and Tackley, 2008). Such 2-D models can also provide a rapid and broad parameter space appraisal, allowing one to focus on a targeted and more sparse study in a full 3-D spherical geometry.

Recent decades have seen extensive validation, verification and benchmarking of Cartesian mantle dynamics codes, in both 2- and 3-D. Verification is typically achieved via comparisons of numerical predictions against analytical solutions (e.g. Zhong

et al., 1993; Kramer et al., 2012), with benchmarking typically undertaken against solutions from other comparable codes, for incompressible (Blankenbach et al., 1989; Travis et al., 1990; Busse et al., 1994; van Keken et al., 1997, 2008; Tosi et al., 2015) and compressible (King et al., 2009) convection. See also Popov et al. (2014) for an overview of geodynamical benchmarks. The number of comparable studies, within a 2-D cylindrical or 3-D spherical geometry, however, is more limited. Given a recent surge in the state-of-the-art tools available for simulating mantle dynamics in these geometries (e.g. Kronbichler et al.,

2012; Logg et al., 2012; Burstedde et al., 2013; Rathgeber et al., 2016; Heister et al., 2017; Wilson et al., 2017), it is important that these tools are verified and validated against a range of analytical and benchmark solutions.

A popular method to obtain analytical solutions is the Method of Manufactured Solutions (MMS; Roache, 2002). With this approach, an arbitrary analytical solution is chosen beforehand and the necessary forcing terms on the right-hand side are derived by substitution of the solution into the left-hand side of the flow equations. A drawback of this approach is that it can

be hard to choose solutions: (i) that satisfy physically realistic boundary conditions, in particular, in less trivial domains; and (ii) with a velocity that is divergence free, to avoid unnatural source terms. Alternatively, one can choose a simple analytical expression for the forcing but the derivation of the corresponding analytical solutions is often laborious.

For the Stokes equations, a well-known set of solutions in the latter category is based on a forcing term in the form of a delta-function, corresponding to an infinitely thin density anomaly at a certain depth. The solutions to these are also used in the

propagator matrix method, where a convolution of delta-function solutions at different depths is used to obtain the response to arbitrary density anomalies (e.g. Hager and Richards, 1989). Solutions for the Cartesian case have been published (e.g. Zhong et al., 1993; Kramer et al., 2012). Spherical solutions can be found in, for example, Ribe (2009). They have previously been used for validation by, for example, Zhong et al. (2000, 2008), Choblet et al. (2007), Davies et al. (2013) and Burstedde et al. (2013). The derivation of these solutions is non-trivial and, often, only cases with simpler free-slip boundary conditions are

explicitly presented. Here, we derive solutions for both free- and zero-slip boundary conditions. In our convergence analyses, where we compare the analytical solutions with numerical solutions obtained using the Fluidity computational modelling framework (Davies et al., 2011), we discuss a limitation of this set of solutions for the benchmarking of geodynamic codes.

For this reason, we also present solutions based on a smooth forcing term, with radial dependence formed by a polynomial of arbitrary order, again for free- and zero-slip cases. This set of solutions provides a flexible way to test mantle dynamics

codes with physically relevant solutions, where, for instance, a high order polynomial forcing can be used to obtain solutions with a strong gradient near the surface. The radial dependence can be combined with spherical harmonics of arbitrary degree and order.

A key step in the derivation of these benchmarks is a decomposition of the solution into poloidal and toroidal components in the Mie representation (Backus, 1986). This results in a biharmonic equation for the polodoidal scalar function. Combined with a set of conditions for the radial dependence of this poloidal function, analytical Stokes solutions can be obtained that satisfy desired free-slip or zero-slip conditions. Similar techniques have been used in Tosi and Martinec (2007) and Horbach et al. (2020). In Section 5.1, we will discuss previously published analytical benchmark cases in shell domains and how they differ from those presented here (e.g. Blinova et al., 2016; Thieulot, 2017; Horbach et al., 2020).

Finally, in addition to the delta-function and smooth cases with either free-slip or zero-slip boundary conditions in a spherical-shell domain, we present the solutions for the corresponding four cases in a 2-D cylindrical shell domain (annulus). Although ultimately, mantle convection is a 3-D phenomenon, a number of processes can be modelled adequately in two dimensions and, accordingly, access to benchmark cases for 2-D numerical models is equally important. The number of published analytical Stokes solutions in 2-D cylindrical shell domains, which are suitable as geodynamical benchmarks and include a complete derivation, is limited. By presenting this extensive set of explicit analytical solutions, we provide a suite of verification cases for use by the wider community of mantle dynamics code developers. An implementation of the solutions is provided through the python package Assess (*Analytical Solutions for the Stokes Equations in Spherical Shells*; Kramer, 2020).

The remainder of this paper is structured as follows. In Section 2 we derive the analytical solutions in cylindrical (section 2.2) and spherical (section 2.3) geometries. Smooth solutions are first provided, for free-slip and zero-slip cases, followed by the delta-function solutions. In Section 3, we briefly describe the Fluidity computational modelling framework, which is used in Section 4 to obtain convergence results of the P2-P1 finite element discretisation to the analytical solutions. In Section 5 we discuss these results, and relate the analytical solutions presented here to those that have previously been published. In particular, we discuss an issue with the delta-function cases for discretisations that use continuous pressure. To demonstrate, we also show results with a $P2_{bubble}$-$P1_{DG}$ discretisation that overcomes this issue.

## 2  Analytical Solutions

### 2.1  Equations

The following derivation is applicable to the incompressible Stokes equations

$$-\nabla \cdot \boldsymbol{\tau} + \nabla p = -g\rho' \hat{\boldsymbol{r}}, \tag{1}$$

$$\boldsymbol{\tau} = \nu \left[ \nabla \boldsymbol{u} + \nabla \boldsymbol{u}^T \right], \tag{2}$$

$$\nabla \cdot \boldsymbol{u} = 0, \tag{3}$$

where the unknowns are velocity $\boldsymbol{u}$ and pressure $p$, with an assumed constant kinematic viscosity $\nu$. The buoyancy force on the right hand side (RHS) of (1) is based on a gravity of constant magnitude $g$, directed in the inward radial direction $-\hat{\boldsymbol{r}}$, and a dimensionless density deviation $\rho'$: $\rho = \rho_0(1 + \rho')$, where $\rho$ is the density and the reference density $\rho_0$ is constant. The equations are solved in a 2-D cylindrical or 3-D spherical domain, bounded by $R_- \leq r \leq R_+$, where $r$ is the radial distance to the origin.

## 2.2 Cylindrical

In 2-D, any incompressible velocity field $\boldsymbol{u}$ can be written as the skew gradient of a streamfunction $\psi$

$$u_r = -\frac{1}{r}\frac{\partial\psi}{\partial\varphi}, \quad u_\varphi = \frac{\partial\psi}{\partial r}, \tag{4}$$

where $\varphi$ is the angle from the $x$-axis in polar coordinates, $x = r\cos(\varphi), y = r\sin(\varphi)$, and $u_r$ and $u_\varphi$ are the radial and transverse components of velocity respectively. The normal deviatoric stress and shear stress components are given by

$$\tau_{rr} = 2\nu\hat{\boldsymbol{r}}\cdot[\nabla\boldsymbol{u}]\cdot\hat{\boldsymbol{r}} = -2\nu\frac{\partial}{\partial r}\left(\frac{1}{r}\frac{\partial\psi}{\partial\varphi}\right), \tag{5}$$

$$\tau_{r\varphi} = \nu\hat{\boldsymbol{r}}\cdot[\nabla\boldsymbol{u}]\cdot\hat{\boldsymbol{\varphi}} + \nu\hat{\boldsymbol{\varphi}}\cdot[\nabla\boldsymbol{u}]\cdot\hat{\boldsymbol{r}} = \nu\left(\frac{\partial^2\psi}{\partial r^2} - \frac{1}{r}\frac{\partial\psi}{\partial r} - \frac{1}{r^2}\frac{\partial^2\psi}{\partial\varphi^2}\right). \tag{6}$$

The momentum equation (1) can be decomposed into radial and transverse components

$$\frac{\nu}{r}\frac{\partial}{\partial\varphi}\nabla^2\psi + \frac{\partial p}{\partial r} = -g\rho', \tag{7}$$

$$-\nu\frac{\partial}{\partial r}\nabla^2\psi + \frac{1}{r}\frac{\partial p}{\partial\varphi} = 0, \tag{8}$$

where the Laplacian, in polar coordinates, is given by

$$\nabla^2\psi = \frac{1}{r}\frac{\partial}{\partial r}\left(r\frac{\partial\psi}{\partial r}\right) + \frac{1}{r^2}\frac{\partial^2\psi}{\partial\varphi^2}. \tag{9}$$

For a derivation of Equations 5–8, see appendix A.

The curl of the momentum equation is obtained by summation of the operators $-\frac{1}{r}\frac{\partial\bullet}{\partial\varphi}$ and $\frac{1}{r}\frac{\partial}{\partial r}(r\bullet)$ applied to the radial (7) and transverse (8) components respectively, which leads to

$$-\nabla^4\psi = \frac{g}{\nu}\frac{1}{r}\frac{\partial\rho'}{\partial\varphi}. \tag{10}$$

The general, real-valued, solution to the biharmonic equation $\nabla^4\psi = 0$ is given by

$$\psi(r,\varphi) = \sum_{n>1}\left(A_n r^n + B_n r^{-n} + C_n r^{n+2} + D_n r^{-n+2}\right)\left(e_n\sin(n\varphi) + f_n\cos(n\varphi)\right)$$

$$+ \left(A_1 r + B_1 r^{-1} + C_1 r^3 + D_1 r\ln r\right)\left(e_1\sin(\varphi) + f_1\cos(\varphi)\right)$$

$$+ A_0 + B_0\ln r + C_0 r^2 + D_0 r^2\ln r, \tag{11}$$

where $A_n$, $B_n$, $C_n$, $D_n$, $e_n$, and $f_n$ are constant coefficients, $n \geq 0$. The $A_n$ and $B_n$ terms are the standard solutions to the homogeneous harmonic equation, and the $C_n$ and $D_n$ terms are obtained as inhomogeneous solutions of the harmonic equation with the homogeneous harmonic solutions ($A_n$ and $B_n$ terms) as the right-hand side. The fact that these (the $C_n$ and $D_n$ terms) are homogeneous biharmonic solutions then follows from: $\nabla^4 = \nabla^2 \nabla^2$. In the following we will, for simplicity, focus on $\sin(n\varphi)$-solutions for a single $n > 1$ and set $e_n = 1, f_n = 0$.

An equation for pressure can be derived by taking the *divergence* of the momentum equation

$$\nabla^2 p = -\frac{g}{r}\frac{\partial (r\rho')}{\partial r}. \tag{12}$$

From (7) it can be seen that $\sin(n\varphi)$ solutions for $\psi$ are associated with $\cos(n\varphi)$ solutions of $p$ and $\rho'$. The homogeneous solutions, i.e. $\rho' = 0$, are thus the standard harmonic solutions (again neglecting the $n = 0, 1$ solutions)

$$p(r,\varphi) = \sum_{n>1} \left( G_n r^n + H_n r^{-n} \right) \cos(n\varphi), \tag{13}$$

where $G_n$ and $H_n$ are constant coefficients, $n > 1$.

After substitution of $p$ and $\psi$ in (7)–(8), the following relations

$$G_n = -4\nu C_n(n+1), \quad H_n = -4\nu D_n(n-1), \tag{14}$$

between the coefficients of the homogeneous solutions for $\psi$ and $p$ can be derived.

### 2.2.1 Smooth density profile – cylindrical

In the first test case we consider a density perturbation of the following form

$$\rho' = \frac{r^k}{R_+^k} \cos(n\varphi) \tag{15}$$

with $k > 0$. It is easily verified that an inhomogeneous solution to (10) exists of the form

$$\psi = E r^{k+3} \sin(n\varphi), \quad E = \frac{g R_+^{-k} n}{\nu \left( (k+3)^2 - n^2 \right) \left( (k+1)^2 - n^2 \right)} \tag{16}$$

assuming $k \neq n - 3$ and $k \neq n - 1$, so that a general solution can be written as

$$\psi(r,\varphi) = \left( A r^n + B r^{-n} + C r^{n+2} + D r^{-n+2} + E r^{k+3} \right) \sin(n\varphi). \tag{17}$$

Note that for the remainder of this derivation we drop the subscript $n$ in the coefficients for $A, B, C$ and $D$.

An inhomogeneous solution for pressure of (12) is given by

$$p(r,\varphi) = F r^{k+1} \cos(n\varphi), \quad F = -\frac{g R_+^{-k}(k+1)}{(k+1)^2 - n^2}. \tag{18}$$

The general solution for pressure is thus

$$p(r,\varphi) = \left( G r^n + H r^{-n} + F r^{k+1} \right) \cos(n\varphi) \tag{19}$$

 with $G$ and $H$ given by (14).

The four remaining coefficients $A, B, C$, and $D$, are fixed by a choice of boundary conditions at the inner and outer surfaces of the cylindrical domain at $r = R_+$ and $r = R_-$, respectively. At both, no-normal flow conditions are imposed via $\frac{\partial \psi}{\partial \varphi} = 0$. Two further equations are found by imposing either $\tau_{r\varphi} = 0$ (free slip), or $\frac{\partial \psi}{\partial r} = 0$ (zero slip) at both boundaries.

The solution coefficients for free-slip, no-normal flow at both boundaries are given by

$$A = \frac{gR_+^{-n+3}}{\nu} \frac{\alpha^{k+n+3} - \alpha^2}{4(\alpha + \alpha^n)(\alpha^n - \alpha)(k+n+1)(k-n+3)}$$

$$B = \frac{gR_+^{n+3}}{\nu} \frac{\alpha^{k+n+3} - \alpha^{2n+2}}{4(\alpha^{n+1} + 1)(\alpha^{n+1} - 1)(k+n+3)(k-n+1)}$$

$$C = \frac{gR_+^{-n+1}}{\nu} \frac{-\alpha^{k+n+3} + 1}{4(\alpha^{n+1} + 1)(\alpha^{n+1} - 1)(k+n+3)(k-n+1)}$$

$$D = \frac{gR_+^{n+1}}{\nu} \frac{-\alpha^{k+n+3} + \alpha^{2n}}{4(\alpha + \alpha^n)(\alpha^n - \alpha)(k+n+1)(k-n+3)}$$

where we use $\alpha = R_-/R_+$.

The zero-slip solution coefficients are given by

$$A = \frac{gR_+^{-n+3}n}{\nu} \frac{\left(\alpha^{k+n+3} + \alpha^{2n}\right)(k+n+1)(n+1) - \left(\alpha^{k+n+1} + \alpha^{2n+2}\right)(k+n+3)n - \left(\alpha^{k+3n+3} + 1\right)(k-n+1)}{2\left(\left(\alpha^{n+1} - \alpha^{n-1}\right)^2 n^2 - \left(\alpha^{2n} - 1\right)^2\right)\left((k+3)^2 - n^2\right)\left((k+1)^2 - n^2\right)}$$

$$B = \frac{gR_+^{n+3}n}{\nu} \frac{-\left(\alpha^{k+3n+3} + \alpha^{2n}\right)(k-n+1)(n-1) + \left(\alpha^{k+3n+1} + \alpha^{2n+2}\right)(k-n+3)n - \left(\alpha^{k+n+3} + \alpha^{4n}\right)(k+n+1)}{2\left(\left(\alpha^{n+1} - \alpha^{n-1}\right)^2 n^2 - \left(\alpha^{2n} - 1\right)^2\right)\left((k+3)^2 - n^2\right)\left((k+1)^2 - n^2\right)}$$

$$C = \frac{gR_+^{-n+1}n}{\nu} \frac{\left(\alpha^{k+n+1} + \alpha^{2n}\right)(k+n+3)(n-1) - \left(\alpha^{k+n+3} + \alpha^{2n-2}\right)(k+n+1)n + \left(\alpha^{k+3n+1} + 1\right)(k-n+3)}{2\left(\left(\alpha^{n+1} - \alpha^{n-1}\right)^2 n^2 - \left(\alpha^{2n} - 1\right)^2\right)\left((k+3)^2 - n^2\right)\left((k+1)^2 - n^2\right)}$$

$$D = \frac{gR_+^{n+1}n}{\nu} \frac{-\left(\alpha^{k+3n+1} + \alpha^{2n}\right)(k-n+3)(n+1) + \left(\alpha^{k+3n+3} + \alpha^{2n-2}\right)(k-n+1)n + \left(\alpha^{k+n+1} + \alpha^{4n}\right)(k+n+3)}{2\left(\left(\alpha^{n+1} - \alpha^{n-1}\right)^2 n^2 - \left(\alpha^{2n} - 1\right)^2\right)\left((k+3)^2 - n^2\right)\left((k+1)^2 - n^2\right)}.$$

### 2.2.2 Green's function solution – cylindrical

Another set of useful solutions is found considering the following perturbation density

$$\rho' = \delta(r - r')\cos(n\varphi) \tag{20}$$

representing an infinitely thin density anomaly at $r = r'$ with $R_- < r' < R_+$. Since $\rho' = 0$ for $r' \neq r$, the solution is described by combining two homogeneous solutions

$$\psi(r, \varphi) = \begin{cases} \psi_-(r, \varphi) = \left(A_- r^n + B_- r^{-n} + C_- r^{n+2} + D_- r^{-n+2}\right)\sin(n\varphi) & \text{for } R_- \leq r < r', \\ \psi_+(r, \varphi) = \left(A_+ r^n + B_+ r^{-n} + C_+ r^{n+2} + D_+ r^{-n+2}\right)\sin(n\varphi) & \text{for } r' < r \leq R_+. \end{cases} \tag{21}$$

We thus have 8 coefficients that are fixed by boundary conditions and conditions at the interface. Boundary conditions at the inner and outer boundary again provide four equations. Furthermore we impose continuity of both velocity components at the

interface

$$\psi_-(r',\varphi) = \psi_+(r',\varphi), \quad \text{and} \quad \frac{\partial \psi_-}{\partial r}(r',\varphi) = \frac{\partial \psi_+}{\partial r}(r',\varphi). \tag{22}$$

Since no lateral force is being applied at the interface we also expect continuity of the shear stress (6) which in combination with the above implies

$$\frac{\partial^2 \psi_-}{\partial r^2}(r',\varphi) = \frac{\partial^2 \psi_+}{\partial r^2}(r',\varphi). \tag{23}$$

Finally the eighth equation is obtained by integrating (10) over a small strip $r' - \epsilon \le r \le r' + \epsilon$, between two arbitrary angles $\varphi_1 \le \varphi \le \varphi_2$:

$$\int_{r=r'-\epsilon}^{r'+\epsilon} \int_{\varphi=\varphi_1}^{\varphi_2} \left(\nabla^4 \psi\right) r \, dr \, d\varphi = \int_{r=r'-\epsilon}^{r'+\epsilon} \int_{\varphi=\varphi_1}^{\varphi_2} \frac{gn\sin(n\varphi)}{\nu r}\delta(r-r') r \, dr \, d\varphi, \tag{24}$$

$$\left[\int_{\varphi=\varphi_1}^{\varphi_2} \hat{\boldsymbol{r}} \cdot \nabla\left(\nabla^2\psi\right) r \, d\varphi\right]_{r=r'-\epsilon}^{r=r'+\epsilon} + \left[\int_{r=r'-\epsilon}^{r'+\epsilon} \hat{\boldsymbol{\varphi}} \cdot \nabla\left(\nabla^2\psi\right) r \, dr\right]_{\varphi=\varphi_1}^{\varphi_2} = \int_{\varphi=\varphi_1}^{\varphi_2} \frac{gn\sin(n\varphi)}{\nu} \, d\varphi. \tag{25}$$

Taking the limit $\epsilon \to 0$, the second term on the left-hand side disappears, whereas the first term becomes

$$\int_{\varphi=\varphi_1}^{\varphi_2} \frac{\partial}{\partial r}\left(\frac{1}{r}\frac{\partial}{\partial r}\left(r\frac{\partial \psi_+}{\partial r}\right) + \frac{1}{r^2}\frac{\partial^2 \psi_+}{\partial \varphi^2}\right)\bigg|_{r=r'} r' \, d\varphi - \int_{\varphi=\varphi_1}^{\varphi_2} \frac{\partial}{\partial r}\left(\frac{1}{r}\frac{\partial}{\partial r}\left(r\frac{\partial \psi_-}{\partial r}\right) + \frac{1}{r^2}\frac{\partial^2 \psi_-}{\partial \varphi^2}\right)\bigg|_{r=r'} r' \, d\varphi.$$

Again using (22) and (23), only the jump term for the third radial derivative of $\psi_\pm$ remains:

$$\int_{\varphi=\varphi_1}^{\varphi_2} \left(\frac{\partial^3 \psi_+}{\partial r^3}(r',\varphi) - \frac{\partial^3 \psi_-}{\partial r^3}(r',\varphi)\right) r' \, d\varphi = \int_{\varphi=\varphi_1}^{\varphi_2} \frac{gn\sin(n\varphi)}{\nu} \, d\varphi.$$

Thus for this to hold for arbitrary $\varphi_1$ and $\varphi_2$, we need

$$\frac{\partial^3 \psi_+}{\partial r^3}(r',\varphi) - \frac{\partial^3 \psi_-}{\partial r^3}(r',\varphi) = \frac{gn\sin(n\varphi)}{\nu r'}. \tag{26}$$

The solution coefficients for free-slip boundary conditions at $r = R_-$ and $r = R_+$ are

$$A_\pm = \frac{gr'^{-n+2}}{\nu} \frac{\pm\left(\alpha_\mp^{2n-2} - 1\right)}{8\left(\alpha_\pm^{2n-2} - \alpha_\mp^{2n-2}\right)(n-1)}$$

$$B_\pm = \frac{gr'^{n+2}}{\nu} \frac{\pm\left(\alpha_\mp^{2n+2} - 1\right)\alpha_\pm^{2n+2}}{8\left(\alpha_\pm^{2n+2} - \alpha_\mp^{2n+2}\right)(n+1)}$$

$$C_\pm = \frac{gr'^{-n}}{\nu} \frac{\pm\left(\alpha_\mp^{2n+2} - 1\right)}{8\left(\alpha_\mp^{2n+2} - \alpha_\pm^{2n+2}\right)(n+1)}$$

$$D_\pm = \frac{gr'^{n}}{\nu} \frac{\pm\left(\alpha_\mp^{2n-2} - 1\right)\alpha_\pm^{2n-2}}{8\left(\alpha_\mp^{2n-2} - \alpha_\pm^{2n-2}\right)(n-1)}$$

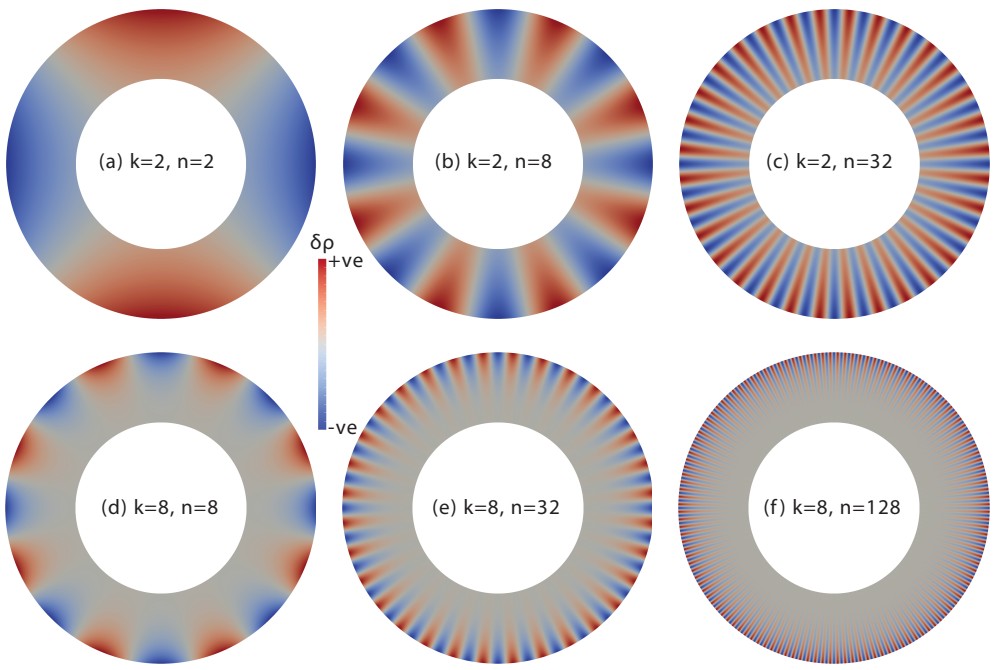

**Figure 1.** Density perturbation ($\delta\rho$) field for smooth cylindrical cases across a range of $n$ and $k$. As the wavenumber $n$ increases, the wavelength of the perturbation decreases. As $k$ increases, the perturbation becomes more concentrated towards the domain's outer boundary.

where we use $\alpha_\pm = R_\pm/r'$.

Zero-slip conditions at both boundaries lead to

$$A_\pm = \frac{gr'^{-n+2}}{\nu}\frac{\left(\left(\alpha_+^2 - \alpha_-^2\right)n + \alpha_+^{-2n} - \alpha_-^{-2n} \pm (n+1)\right)(n-1) - \left(\alpha_+^{-2n}\alpha_-^2 - \alpha_+^2\alpha_-^{-2n}\right)n \pm \left(\gamma^{\pm 2n} - \gamma^{\mp 2}n^2\right)}{8\left((\gamma - \gamma^{-1})^2 n^2 - (\gamma^{-n} - \gamma^n)^2\right)(n-1)}$$

$$B_\pm = \frac{gr'^{n+2}}{\nu}\frac{\left(\left(\alpha_+^2 - \alpha_-^2\right)n - \alpha_+^{2n} + \alpha_-^{2n} \pm (n-1)\right)(n+1) + \left(\alpha_+^{2n}\alpha_-^2 - \alpha_+^2\alpha_-^{2n}\right)n \pm \left(\gamma^{\mp 2n} - \gamma^{\mp 2}n^2\right)}{8\left((\gamma - \gamma^{-1})^2 n^2 - (\gamma^{-n} - \gamma^n)^2\right)(n+1)}$$

$$C_\pm = \frac{gr'^{-n}}{\nu}\frac{\left(\left(\alpha_-^{-2} - \alpha_+^{-2}\right)n + \alpha_+^{-2n} - \alpha_-^{-2n} \mp (n-1)\right)(n+1) - \left(\alpha_+^{-2n}\alpha_-^{-2} - \alpha_+^{-2}\alpha_-^{-2n}\right)n \mp \left(\gamma^{\pm 2n} - \gamma^{\pm 2}n^2\right)}{8\left((\gamma - \gamma^{-1})^2 n^2 - (\gamma^{-n} - \gamma^n)^2\right)(n+1)}$$

$$D_\pm = \frac{gr'^{n}}{\nu}\frac{\left(\left(\alpha_-^{-2} - \alpha_+^{-2}\right)n - \alpha_+^{2n} + \alpha_-^{2n} \mp (n+1)\right)(n-1) + \left(\alpha_+^{2n}\alpha_-^{-2} - \alpha_+^{-2}\alpha_-^{2n}\right)n \mp \left(\gamma^{\mp 2n} - \gamma^{\pm 2}n^2\right)}{8\left((\gamma - \gamma^{-1})^2 n^2 - (\gamma^{-n} - \gamma^n)^2\right)(n-1)}$$

where in addition we use $\gamma = R_-/R_+$.

## 2.3 Spherical

In this section we derive the equivalent of the four cylindrical cases in a 3-D spherical domain, $R_- \leq r \leq R_+$. A more detailed derivation of the equations can be found in Ribe (2009). In 3-D, a solenoidal velocity field $\boldsymbol{u}$ can be decomposed as

$$\boldsymbol{u} = \nabla \times (\boldsymbol{r} \times \nabla \mathcal{P}) + \boldsymbol{r} \times \nabla \mathcal{T} \tag{27}$$

using poloidal and toroidal scalar fields $\mathcal{P}$ and $\mathcal{T}$ (Backus, 1986), where $\boldsymbol{r} = r\hat{\boldsymbol{r}}$. In spherical coordinates

$$u_r = \frac{1}{r}\Lambda^2 \mathcal{P}, \quad u_\theta = -\frac{1}{r}\frac{\partial^2 (r\mathcal{P})}{\partial r \partial \theta} - \frac{1}{\sin(\theta)}\frac{\partial \mathcal{T}}{\partial \varphi}, \quad u_\varphi = -\frac{1}{r\sin(\theta)}\frac{\partial^2 (r\mathcal{P})}{\partial r \partial \varphi} + \frac{\partial \mathcal{T}}{\partial \theta} \tag{28}$$

where $\varphi$ is the longitude and $\theta$ is the co-latitude, and

$$\Lambda^2 \mathcal{P} = \frac{1}{\sin(\theta)}\frac{\partial}{\partial \theta}\left(\sin(\theta)\frac{\partial \mathcal{P}}{\partial \theta}\right) + \frac{1}{\sin(\theta)^2}\frac{\partial^2 \mathcal{P}}{\partial \varphi^2}, \text{ so that } \nabla^2 = \frac{1}{r^2}\left(\frac{\partial}{\partial r}r^2\frac{\partial}{\partial r} + \Lambda^2\right). \tag{29}$$

We further derive the components of stress normal to a spherical surface

$$\tau_{rr} = 2\nu \frac{\partial u_r}{\partial r} = 2\nu \frac{\partial}{\partial r}\left(\frac{1}{r}\Lambda^2 \mathcal{P}\right), \tag{30}$$

$$\tau_{r\theta} = \nu r \frac{\partial}{\partial r}\left(\frac{u_\theta}{r}\right) + \frac{\nu}{r}\frac{\partial u_r}{\partial \theta} = \nu \frac{\partial}{\partial \theta}\left(\frac{1}{r^2}\Lambda^2 \mathcal{P} - \frac{\partial^2 \mathcal{P}}{\partial r^2} + \frac{2}{r^2}\mathcal{P}\right) - \nu r \frac{\partial}{\partial r}\left(\frac{1}{r\sin(\theta)}\frac{\partial \mathcal{T}}{\partial \varphi}\right), \tag{31}$$

$$\tau_{r\varphi} = \nu r \frac{\partial}{\partial r}\left(\frac{u_\varphi}{r}\right) + \frac{\nu}{r\sin(\theta)}\frac{\partial u_r}{\partial \varphi} = \frac{\nu}{\sin(\theta)}\frac{\partial}{\partial \varphi}\left(\frac{1}{r^2}\Lambda^2 \mathcal{P} - \frac{\partial^2 \mathcal{P}}{\partial r^2} + \frac{2}{r^2}\mathcal{P}\right) + \nu r \frac{\partial}{\partial r}\left(\frac{1}{r}\frac{\partial \mathcal{T}}{\partial \theta}\right). \tag{32}$$

Using these, we can work out spherical components of the momentum equation (1) to be

$$-\frac{\nu}{r}\Lambda^2 \nabla^2 \mathcal{P} + \frac{\partial p}{\partial r} = -g\rho', \tag{33}$$

$$\frac{\nu}{r}\frac{\partial^2 (r\nabla^2 \mathcal{P})}{\partial r \partial \theta} + \frac{\nu}{\sin(\theta)}\frac{\partial \nabla^2 \mathcal{T}}{\partial \varphi} + \frac{1}{r}\frac{\partial p}{\partial \theta} = 0, \tag{34}$$

$$\frac{\nu}{r\sin(\theta)}\frac{\partial^2 (r\nabla^2 \mathcal{P})}{\partial r \partial \varphi} + \nu\frac{\partial \nabla^2 \mathcal{T}}{\partial \theta} + \frac{1}{r\sin(\theta)}\frac{\partial p}{\partial \varphi} = 0. \tag{35}$$

As can be seen from these equations, the toroidal part of the solution is independent of the density distribution. A nonzero toroidal component is only introduced through a particular choice of boundary conditions. For the test cases in this paper we only consider free-slip and zero-slip boundary conditions with radial forcing so we may assume $\mathcal{T} = 0$. The assumption of the velocity field being solenoidal, which underlies the Mie representation (27), follows directly from the incompressibility condition and the no-normal flow condition on the boundary.

Taking the curl of the momentum equation

$$-\hat{\boldsymbol{\theta}}\frac{\nu}{\sin(\theta)}\frac{\partial \nabla^4 \mathcal{P}}{\partial \varphi} + \hat{\boldsymbol{\varphi}}\nu\frac{\partial \nabla^4 \mathcal{P}}{\partial \theta} = -\hat{\boldsymbol{\theta}}\frac{1}{r\sin(\theta)}\frac{\partial g\rho'}{\partial \varphi} + \hat{\boldsymbol{\varphi}}\frac{1}{r}\frac{\partial g\rho'}{\partial \theta}. \tag{36}$$

In a spherical domain this implies that $\nu\nabla^4 \mathcal{P} - g\rho'/r$ varies in the radial direction only. Without loss of generality we may therefore seek solutions $\mathcal{P}$ that satisfy

$$\nabla^4 \mathcal{P} = \frac{g\rho'}{\nu r}, \tag{37}$$

since for any other solution $\mathcal{P} + \mathcal{P}'$ of (36) we know that $\nabla^4 \mathcal{P}'$ is purely radial, and therefore $\mathcal{P}'$ can be written as a sum of biharmonic solutions and a purely radial function. The biharmonic solutions are included in (37) and the purely radial function is discarded by (27).

Solutions to the biharmonic equation $\nabla^4 \mathcal{P} = 0$ in 3-D are given by

$$\mathcal{P}(r,\theta,\varphi) = \sum_{l \geq 0} \sum_{m=-l}^{l} \left( A_{lm} r^l + B_{lm} r^{-l-1} + C_{lm} r^{l+2} + D_{lm} r^{-l+1} \right) Y_{lm}(\theta,\varphi), \tag{38}$$

where $A_{lm}$, $B_{lm}$, $C_{lm}$, and $D_{lm}$ are constant coefficients, and $Y_{lm}$ are Laplace's spherical harmonic functions of degree $l$ and order $m$, which have the property that

$$\Lambda^2 Y_{lm} = -l(l+1) Y_{lm}. \tag{39}$$

As in 2-D, an equation for the pressure is obtained by taking the divergence of the momentum equation, giving

$$\nabla^2 p = -\frac{g}{r^2} \frac{\partial}{\partial r} \left( r^2 \rho' \right). \tag{40}$$

Homogeneous solutions for pressure are written

$$p(r,\theta,\varphi) = \sum_{l \geq 0} \sum_{m=-l}^{l} \left( G_{lm} r^l + H_{lm} r^{-l-1} \right) Y_{lm}(\theta,\varphi), \tag{41}$$

where substitution of (38) and (41) in (33) gives

$$G_{lm} = -2\nu(l+1)(2l+3) C_{lm}, \quad H_{lm} = -2\nu l(2l-1) D_{lm}. \tag{42}$$

### 2.3.1 Smooth density profile – spherical

We consider a density perturbation of the following form

$$\rho' = \frac{r^k}{R_+^k} Y_{lm}(\theta,\varphi). \tag{43}$$

An inhomogeneous solution of (37) is given by

$$\mathcal{P} = E r^{k+3} Y_{lm}, \quad E = \frac{g R_+^{-k}}{\nu\left((k+1)(k+2) - l(l+1)\right)\left((k+3)(k+4) - l(l+1)\right)} \tag{44}$$

with a more generic solution written as

$$\mathcal{P} = \left( A r^l + B r^{-l-1} + C r^{l+2} + D r^{-l+1} + E r^{k+3} \right) Y_{lm} \tag{45}$$

where we have dropped the $lm$ subscripts of the coefficients $A$, $B$, $C$, $D$. The pressure solution for (40) is

$$p = \left( G r^l + H r^{-l-1} + F r^{k+1} \right) Y_{lm}, \quad F = -\frac{g(k+2) R_+^{-k}}{(k+1)(k+2) - l(l+1)}, \tag{46}$$

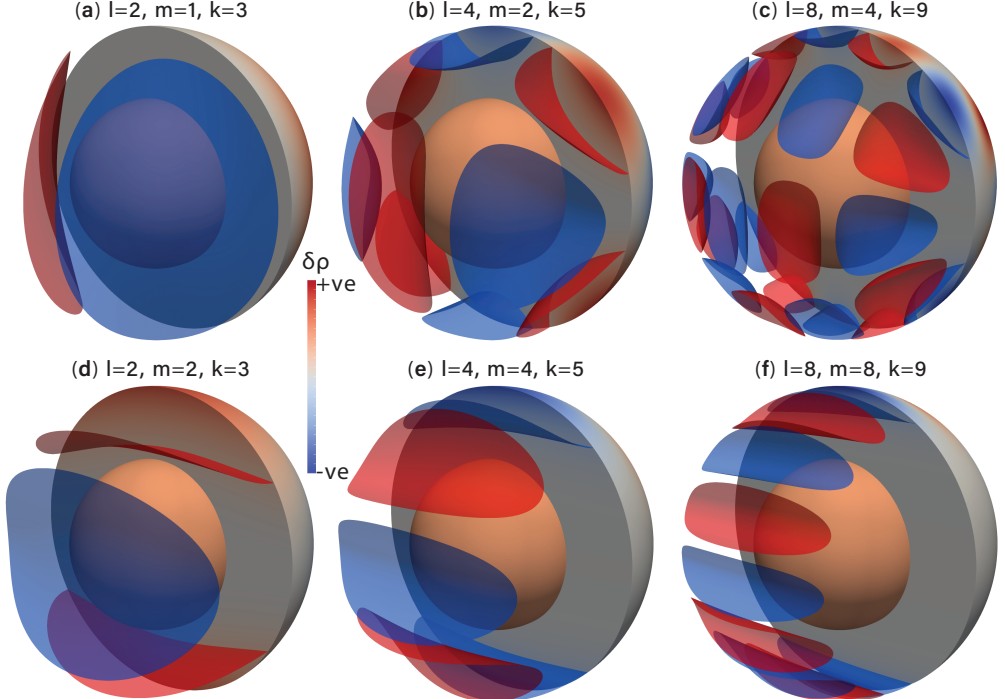

**Figure 2.** Illustrations of the density perturbation ($\delta\rho$) field for smooth spherical cases across a range of $l$, $m$ and $k$.

and $G$ and $H$ are given by (42).

As before, the four coefficients $A$, $B$, $C$, and $D$ are fixed by the boundary conditions

*no-normal flow:*
$$u_r = \frac{1}{r}\Lambda^2\mathcal{P} = 0 \implies \mathcal{P} = 0, \qquad \text{at } r = R_\pm \qquad (47)$$

where we use $\Lambda^2\mathcal{P} = -l(l+1)\mathcal{P}$. The two no-normal flow conditions at $r = R_-$ and $r = R_+$ are combined with two further conditions

*free slip:*
$$\tau_{r\theta} = \tau_{r\varphi} = 0 \implies \frac{1}{r}\Lambda^2\mathcal{P} - \frac{\partial^2\mathcal{P}}{\partial r^2} + \frac{2}{r^2}\mathcal{P} = 0 \implies \frac{\partial^2\mathcal{P}}{\partial r^2} = 0, \qquad \text{at } r = R_\pm, \text{ or} \qquad (48)$$

*zero slip:*
$$u_\theta = u_\varphi = 0 \implies \frac{\partial(r\mathcal{P})}{\partial r} = 0 \implies \frac{\partial\mathcal{P}}{\partial r} = 0, \qquad \text{at } r = R_\pm. \qquad (49)$$

For free-slip conditions at both $r = R_-$ and $r = R_+$, the solution coefficients are given by

$$A = \frac{gR_+^{-l+3}}{\nu} \frac{\alpha^{k+3} - \alpha^{-l+1}}{2\left(\alpha^l - \alpha^{-l+1}\right)(k+l+2)(k-l+3)(2l+1)}$$

$$B = \frac{gR_+^{l+4}}{\nu} \frac{-\alpha^{k+4} + \alpha^{l+3}}{2\left(\alpha^{-l} - \alpha^{l+3}\right)(k+l+4)(k-l+1)(2l+1)}$$

$$C = \frac{gR_+^{-l+1}}{\nu} \frac{\alpha^{k+4} - \alpha^{-l}}{2\left(\alpha^{-l} - \alpha^{l+3}\right)(k+l+4)(k-l+1)(2l+1)}$$

$$D = \frac{gR_+^{l+2}}{\nu} \frac{-\alpha^{k+3} + \alpha^l}{2\left(\alpha^l - \alpha^{-l+1}\right)(k+l+2)(k-l+3)(2l+1)}.$$

Zero-slip conditions at both boundaries lead to the following solution coefficients

$$A = \frac{gR_+^{-l+3}}{\nu} \frac{\left(\alpha^{k+2} + \alpha^{l-1}\right)(k+l+2)(2l+3) - \left(\alpha^k + \alpha^{l+1}\right)(k+l+4)(2l+1) - 2\left(\alpha^{k+2l+3} + \alpha^{-l-2}\right)(k-l+1)}{\Gamma}$$

$$B = \frac{gR_+^{l+4}}{\nu} \frac{\left(\alpha^{k+2l+1} + \alpha^{l+1}\right)(k-l+3)(2l+1) - \left(\alpha^{k+2l+3} + \alpha^{l-1}\right)(k-l+1)(2l-1) - 2\left(\alpha^{k+2} + \alpha^{3l}\right)(k+l+2)}{\Gamma}$$

$$C = \frac{gR_+^{-l+1}}{\nu} \frac{-\left(\alpha^{k+2} + \alpha^{l-3}\right)(k+l+2)(2l+1) + \left(\alpha^k + \alpha^{l-1}\right)(k+l+4)(2l-1) + 2\left(\alpha^{k+2l+1} + \alpha^{-l-2}\right)(k-l+3)}{\Gamma}$$

$$D = \frac{gR_+^{l+2}}{\nu} \frac{-\left(\alpha^{k+2l+1} + \alpha^{l-1}\right)(k-l+3)(2l+3) + \left(\alpha^{k+2l+3} + \alpha^{l-3}\right)(k-l+1)(2l+1) + 2\left(\alpha^k + \alpha^{3l}\right)(k+l+4)}{\Gamma}$$

$$\Gamma = \left(\left(\alpha^{l+1} + \alpha^{l-3}\right)(2l+1)^2 - 2\alpha^{l-1}(2l+3)(2l-1) - 4\alpha^{3l} - 4\alpha^{-l-2}\right)(k+l+4)(k+l+2)(k-l+3)(k-l+1).$$

### 2.3.2 Green's function solution – spherical

As in two dimensions, we find solutions for the case where

$$\rho' = \delta(r - r')Y_{lm}(\theta,\varphi), \tag{50}$$

by combining two homogeneous solutions

$$\mathcal{P}(r,\theta,\varphi) = \begin{cases} \mathcal{P}_-(r,\theta,\varphi) = \left(A_-r^l + B_-r^{-l-1} + C_-r^{l+2} + D_-r^{-l+1}\right)Y_{lm}(\theta,\varphi) & \text{for } R_- \le r < r', \\ \mathcal{P}_+(r,\theta,\varphi) = \left(A_+r^l + B_+r^{-l-1} + C_+r^{l+2} + D_+r^{-l+1}\right)Y_{lm}(\theta,\varphi) & \text{for } r' < r \le R_+. \end{cases} \tag{51}$$

The eight coefficients are found by imposing the same 4 constraints derived from the boundary conditions at $r = R_-$ and $r = R_+$ as in the previous section, and by imposing a further 4 conditions: continuity of all components of $\boldsymbol{u}$, no shear force between the two halves of the domain, and a normal force that is proportional to the density anomaly

*continuity of $u_r$:* $\qquad\qquad\qquad\qquad\qquad \mathcal{P}_-(r',\theta,\varphi) = \mathcal{P}_+(r',\theta,\varphi),$ $\qquad\qquad\qquad$ (52)

*continuity of $u_\theta$ and $u_\varphi$:* $\qquad \dfrac{\partial(r\mathcal{P}_-)}{\partial r}\Big|_{r=r'} = \dfrac{\partial(r\mathcal{P}_+)}{\partial r}\Big|_{r=r'} \implies \dfrac{\partial\mathcal{P}_-}{\partial r}(r',\theta,\varphi) = \dfrac{\partial\mathcal{P}_+}{\partial r}(r',\theta,\varphi),$ (53)

*zero-shear condition:* $\qquad\qquad\qquad \dfrac{\partial^2\mathcal{P}_-}{\partial r^2}(r',\theta,\varphi) = \dfrac{\partial^2\mathcal{P}_+}{\partial r^2}(r',\theta,\varphi),$ $\qquad\qquad$ (54)

*normal-shear condition:* $\qquad\qquad \dfrac{\partial^3\mathcal{P}_+}{\partial r^3}(r',\theta,\varphi) - \dfrac{\partial^3\mathcal{P}_-}{\partial r^3}(r',\theta,\varphi) = \dfrac{gY_{lm}(\theta,\varphi)}{\nu r'},$ $\qquad$ (55)

where (55) is derived from (37) in the same way as (26), and (53)–(55) assume (52).

The free-slip solution coefficients are given by

$$A_\pm = \frac{gr'^{-l+2}}{\nu}\frac{\pm\left(\alpha_\mp^{2l-1}-1\right)}{2\left(\alpha_\pm^{2l-1}-\alpha_\mp^{2l-1}\right)(2l-1)(2l+1)}$$

$$B_\pm = \frac{gr'^{l+3}}{\nu}\frac{\pm\left(\alpha_\mp^{-2l-3}-1\right)}{2\left(\alpha_\pm^{-2l-3}-\alpha_\mp^{-2l-3}\right)(2l+1)(2l+3)}$$

$$C_\pm = \frac{gr'^{-l}}{\nu}\frac{\pm\left(-\alpha_\mp^{2l+3}+1\right)}{2\left(\alpha_\pm^{2l+3}-\alpha_\mp^{2l+3}\right)(2l+1)(2l+3)}$$

$$D_\pm = \frac{gr'^{l+1}}{\nu}\frac{\pm\left(-\alpha_\mp^{-2l+1}+1\right)}{2\left(\alpha_\pm^{-2l+1}-\alpha_\mp^{-2l+1}\right)(2l-1)(2l+1)}.$$

The zero-slip solution coefficients are given by

$$A_\pm = \frac{gr'^{-l+2}}{\nu}\frac{\alpha_+^2-\alpha_-^2+\frac{2}{2l+1}\left(\alpha_+^{-2l-1}-\alpha_-^{-2l-1}\right)\pm\frac{2l+3}{2l+1}+\frac{2}{2l-1}\left(\alpha_+^2\alpha_-^{-2l-1}-\alpha_+^{-2l-1}\alpha_-^2\right)\pm\frac{4\gamma^{\pm(2l+1)}}{(2l-1)(2l+1)}\mp\frac{\gamma^{\mp 2}(2l+1)}{2l-1}}{-8\gamma^{-2l-1}-8\gamma^{2l+1}+2\left(\gamma^2+\gamma^{-2}\right)(2l+1)^2-4(2l-1)(2l+3)}$$

$$B_\pm = \frac{gr'^{l+3}}{\nu}\frac{\alpha_+^2-\alpha_-^2-\frac{2}{2l+1}\left(\alpha_+^{2l+1}-\alpha_-^{2l+1}\right)\pm\frac{2l-1}{2l+1}-\frac{2}{2l+3}\left(\alpha_+^2\alpha_-^{2l+1}-\alpha_+^{2l+1}\alpha_-^2\right)\pm\frac{4\gamma^{\mp(2l+1)}}{(2l+1)(2l+3)}\mp\frac{\gamma^{\mp 2}(2l+1)}{2l+3}}{-8\gamma^{-2l-1}-8\gamma^{2l+1}+2\left(\gamma^2+\gamma^{-2}\right)(2l+1)^2-4(2l-1)(2l+3)}$$

$$C_\pm = \frac{gr'^{-l}}{\nu}\frac{\alpha_-^{-2}-\alpha_+^{-2}+\frac{2}{2l+1}\left(\alpha_+^{-2l-1}-\alpha_-^{-2l-1}\right)\mp\frac{2l-1}{2l+1}+\frac{2}{2l+3}\left(\alpha_+^{-2}\alpha_-^{-2l-1}-\alpha_+^{-2l-1}\alpha_-^{-2}\right)\mp\frac{4\gamma^{\pm(2l+1)}}{(2l+1)(2l+3)}\pm\frac{\gamma^{\pm 2}(2l+1)}{2l+3}}{-8\gamma^{-2l-1}-8\gamma^{2l+1}+2\left(\gamma^2+\gamma^{-2}\right)(2l+1)^2-4(2l-1)(2l+3)}$$

$$D_\pm = \frac{gr'^{l+1}}{\nu}\frac{\alpha_-^{-2}-\alpha_+^{-2}-\frac{2}{2l+1}\left(\alpha_+^{2l+1}-\alpha_-^{2l+1}\right)\mp\frac{2l+3}{2l+1}-\frac{2}{2l-1}\left(\alpha_+^{-2}\alpha_-^{2l+1}-\alpha_+^{2l+1}\alpha_-^{-2}\right)\mp\frac{4\gamma^{\mp(2l+1)}}{(2l-1)(2l+1)}\pm\frac{\gamma^{\pm 2}(2l+1)}{2l-1}}{-8\gamma^{-2l-1}-8\gamma^{2l+1}+2\left(\gamma^2+\gamma^{-2}\right)(2l+1)^2-4(2l-1)(2l+3)}.$$

# 3 Fluidity

The test cases in the previous section have been examined using Fluidity, a finite element, control-volume computational modelling framework (Davies et al., 2011; Kramer et al., 2012).

## 3.1 Discretisation

The numerical solutions for velocity and pressure, $\boldsymbol{u}$ and $p$, are written as a linear combination of basis functions $N_j$ and $M_l$

$$\boldsymbol{u} = \sum_j \boldsymbol{u}_j N_j, \quad p = \sum_l p_l M_l. \tag{56}$$

We use either the P2-P1 (Taylor Hood) or P2$_{\text{bubble}}$-P1$_{\text{DG}}$ element pairs. In both cases the velocity and pressure basis functions $N_j$ and $M_l$ are piecewise quadratic and linear respectively on a triangular (2-D) or tetrahedral (3-D) mesh of the domain $\Omega$. Because the curved boundaries of the cylindrical and spherical-shell domains can only be approximated by the mesh, the numerical domain is denoted by $\Omega_h$. When using the P2-P1 element pair the basis functions are continuous between cells. For the P2$_{\text{bubble}}$-P1$_{\text{DG}}$ element pair the piecewise linear pressure is treated as discontinuous between cells and the continuous

quadratic velocity basis functions are enriched by an extra cubic "bubble" function with a corresponding cell-centred degree of freedom (Ern and Guermond, 2004; Boffi et al., 2013).

The Stokes equations are written in the weak form, using the same $N_j$ and $M_l$ basis as test-functions. After integrating by parts (1) and (3) then become (omitting boundary terms)

$$\int_{\Omega_h} \nu \left( \nabla N_i \right) \cdot \sum_j \left[ \nabla N_j^T \boldsymbol{u}_j + \boldsymbol{u}_j^T \nabla N_j \right] + N_i \sum_l p_l \nabla M_l = - \int_{\Omega_h} N_i g \rho' \hat{\boldsymbol{r}} \qquad \text{for all } N_i, \tag{57}$$

$$\int_{\Omega_h} \left( \nabla M_k \right) \cdot \sum_j \boldsymbol{u}_j N_j = 0 \qquad \text{for all } M_k. \tag{58}$$

Note that we apply strong Dirichlet boundary conditions, so that the boundary integrals can indeed be neglected. In the free-slip case, a local rotation is applied to the velocity vectors, so that the degrees of freedom correspond to velocity components in either the normal or tangential directions. This allows us to enforce a zero normal component, while leaving the tangential components free. For additional details about Fluidity and its implementation see Davies et al. (2011).

### 3.2 Isoparametric representation of the domain

For an accurate representation of the quadratic approximation of velocity, we need to also approximate the curved cylindrical/spherical domain quadratically. This means that rather than each cell in the mesh being described by a linear map $X_{\text{lin}} : \xi \mapsto X_{\text{lin}}(\xi)$ from local coordinates $\xi$ in a reference element to physical coordinates $X$, we use a quadratic map $X_{\text{quad}}(\xi)$, which maps to a curved triangle/tetrahedron that better represents the domain. This map can be obtained from a linear mesh with coordinate mappings, $X_{\text{lin}}$, through quadratic interpolation

$$X_{\text{quad}}(\xi) \overset{\text{quad. interp.}}{\approx} \frac{r_{\text{lin}}(\xi)}{\|X_{\text{lin}}(\xi)\|} X_{\text{lin}}(\xi), \tag{59}$$

at the standard Lagrange nodes of the quadratic function space. Here $r_{\text{lin}}(\xi)$ is the linear interpolation of the radius, i.e. $r_{\text{lin}} = \|X_{\text{lin}}\|$ at the vertices of the linear mesh. This particular choice ensures that for an equal-radius boundary, with the boundary vertices of the linear mesh exactly on the boundary, the quadratic Lagrange nodes also lie exactly on this boundary.

### 3.3 Forcing term

The density perturbation $\rho'$ on the RHS of (57) is a prescribed analytical expression in each of the test cases. For the Green's function solutions in 2-D and 3-D, using (20) and (50) respectively, we get

$$- \int_{\Omega_h} N_i g \rho' \hat{\boldsymbol{r}} = - \int_{\Omega_h} N_i(r, \theta) g \cos(n\theta) \delta(r - r') \hat{\boldsymbol{r}} = - \int_{\Gamma'} N_i(r, \theta) g \cos(n\theta) \boldsymbol{n}, \tag{60}$$

$$- \int_{\Omega_h} N_i(r, \theta) g \rho' \hat{\boldsymbol{r}} = - \int_{\Omega_h} N_i(r, \theta, \varphi) g Y_{lm}(\theta, \varphi) \delta(r - r') \hat{\boldsymbol{r}} = - \int_{\Gamma'} N_i(r, \theta, \varphi) g Y_{lm}(\theta, \varphi) \boldsymbol{n}, \tag{61}$$

where $\Gamma'$ is an internal boundary at $r = r'$ oriented such that its normal $\boldsymbol{n} = \hat{\boldsymbol{r}}$ points outwards.

## 3.4 Solving the linear system and dealing with nullspaces

Equations (57) and (58) form a saddle point linear system which is solved by applying a Schur decomposition technique where the outer iteration, which solves for the pressure degrees of freedom, is solved with a flexible Krylov subspace method, `FGMRES`. The inner solve associated with the velocity degrees of freedom, is solved with the Conjugate Gradient method preconditioned with an algebraic multigrid method (`GAMG` available through PETSc: Balay et al., 1997).

In all cases, the pressure solution is only defined up to an arbitrary constant. The analytical pressure solution has the property that its mean is zero. For comparative purposes we therefore subtract the volume averaged pressure from the obtained numerical pressure solution

$$p \to p - \frac{\int_{\Omega_h} p}{\int_{\Omega_h} 1}. \tag{62}$$

Similarly, for free-slip cases, in 2-D, we may add an arbitrary rotation of the form $(-y, x) = r\hat{\boldsymbol{\theta}}$ to the velocity solution. We therefore apply the following projection to the numerical solution

$$\boldsymbol{u} \to \boldsymbol{u} - \frac{\int_{\Omega_h} r\hat{\boldsymbol{\theta}} \cdot \boldsymbol{u}}{\int_{\Omega_h} r^2} r\hat{\boldsymbol{\theta}}, \tag{63}$$

which ensures that the angular momentum $\int r\hat{\boldsymbol{\theta}} \cdot \boldsymbol{u}$ is zero, as it is for the analytical solutions. In the same way, in three dimensions we subtract the three rotational (rigid body) modes.

It should be noted that the same velocity and pressure modes lead to zero modes (eigenvectors) for the linear system based on (57) and (58), rendering the resulting matrix singular. In preconditioned Krylov methods we typically need to subtract the zero modes from the approximate solution at every iteration. With iterative approximation $\boldsymbol{x}^i$ and zero eigenvector $\boldsymbol{\lambda}$ we get

$$\boldsymbol{x}^i \to \boldsymbol{x}^i - \langle \boldsymbol{\lambda}, \boldsymbol{x}^i \rangle \boldsymbol{\lambda}. \tag{64}$$

This $l_2$-projection, based on the $l_2$ inner product $\langle \cdot, \cdot \rangle$, is analogous but not equivalent to the projections in (62) and (63) (which are $L_2$-projections). Therefore, despite the $l_2$ projections fixing the nullmodes during the iterative solve, the $L_2$-projections should be applied as an additional step after the iterative solvers have completed to ensure convergence to the analytical solution.

## 4 Convergence Results

In this section we show the convergence of the numerical solutions obtained with Fluidity, using the P2-P1 element pair, towards the analytical solutions. For 2-D cylindrical cases, the series of meshes start at refinement level 1, where the mesh consists of 128 divisions in the horizontal, and 16 layers, giving $128 \times 16 \times 2 = 4096$ triangles. At each subsequent level the mesh is refined, doubling the resolution in both directions. For the spherical cases, the mesh at refinement level 1 is obtained from an icosahedron refined three times, starting with 1280 triangles in the horizontal, which is extruded radially to 16 layers,

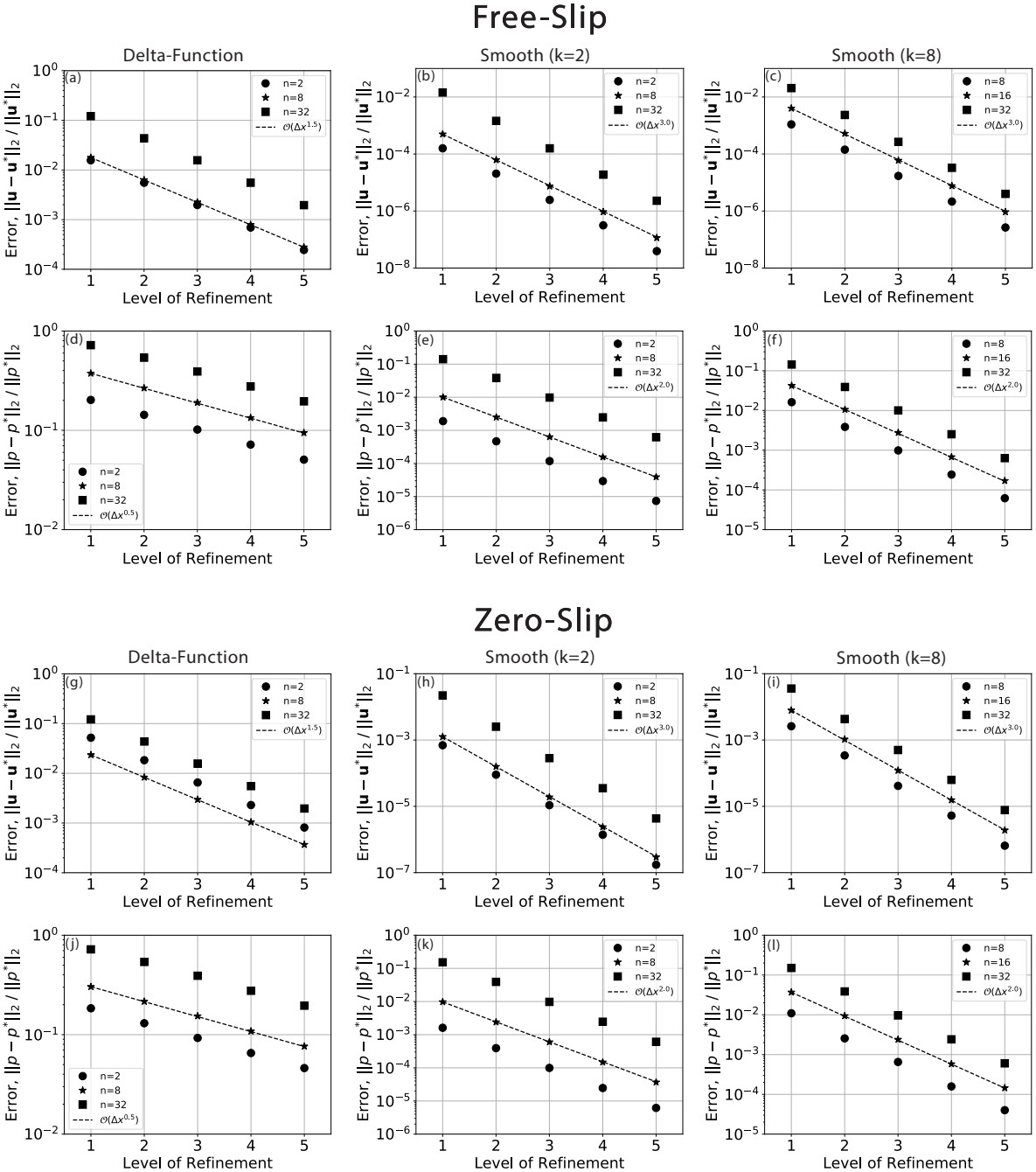

**Figure 3.** Convergence for 2-D cylindrical cases with free-slip and zero-slip boundary conditions, at a series of different wavenumbers, $n$, as indicated in the legend. Note that cases with a smooth forcing are run at $k = 2$ and $k = 8$, as indicated. Convergence rate is indicated by dashed lines, with the order of convergence provided in the legend.

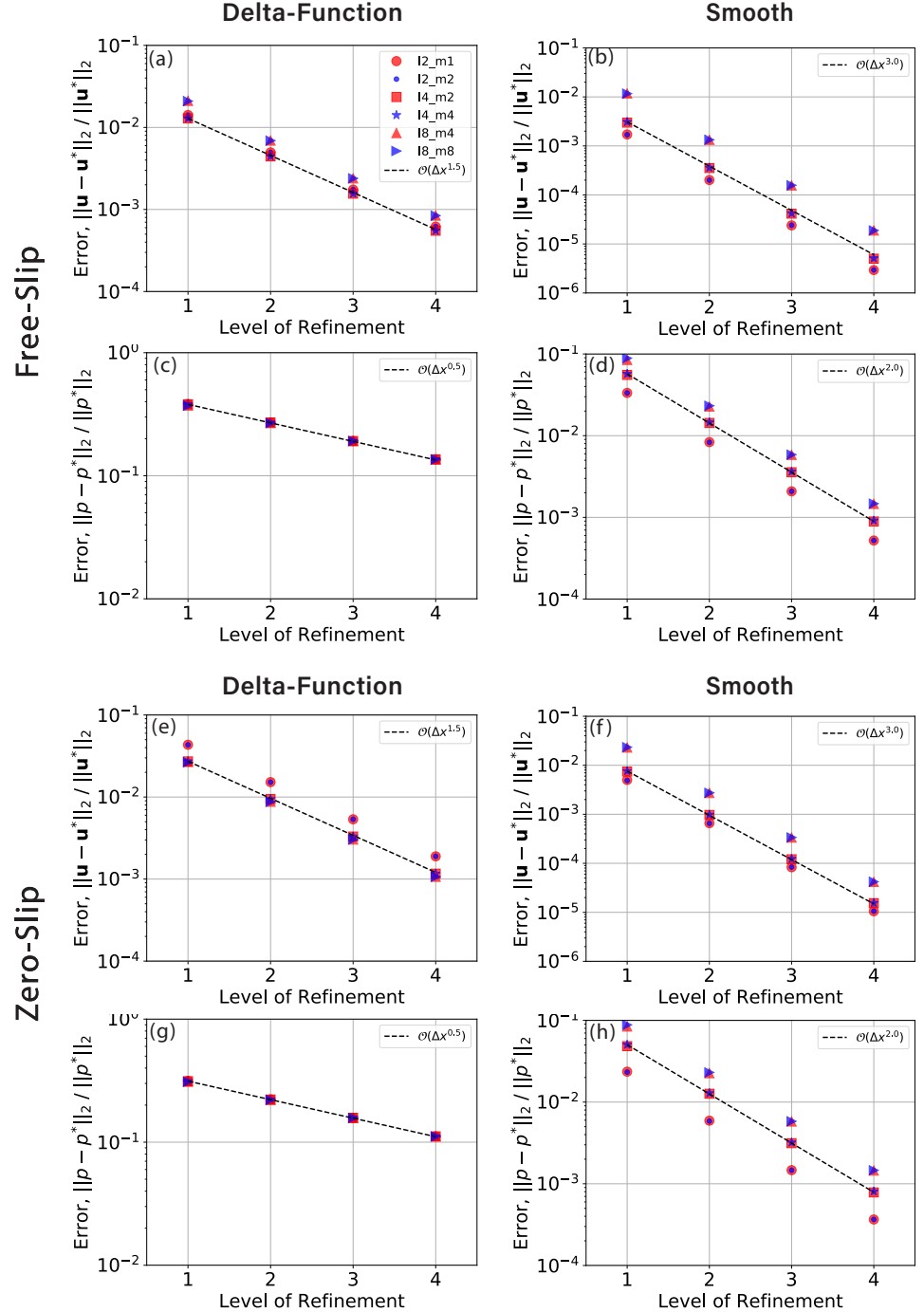

**Figure 4.** Convergence of velocity and pressure for 3-D spherical cases with free-slip and zero-slip boundary conditions, at a range of degrees $l$ and orders $m$. Note that all cases with a smooth forcing are run at $k = l + 1$.

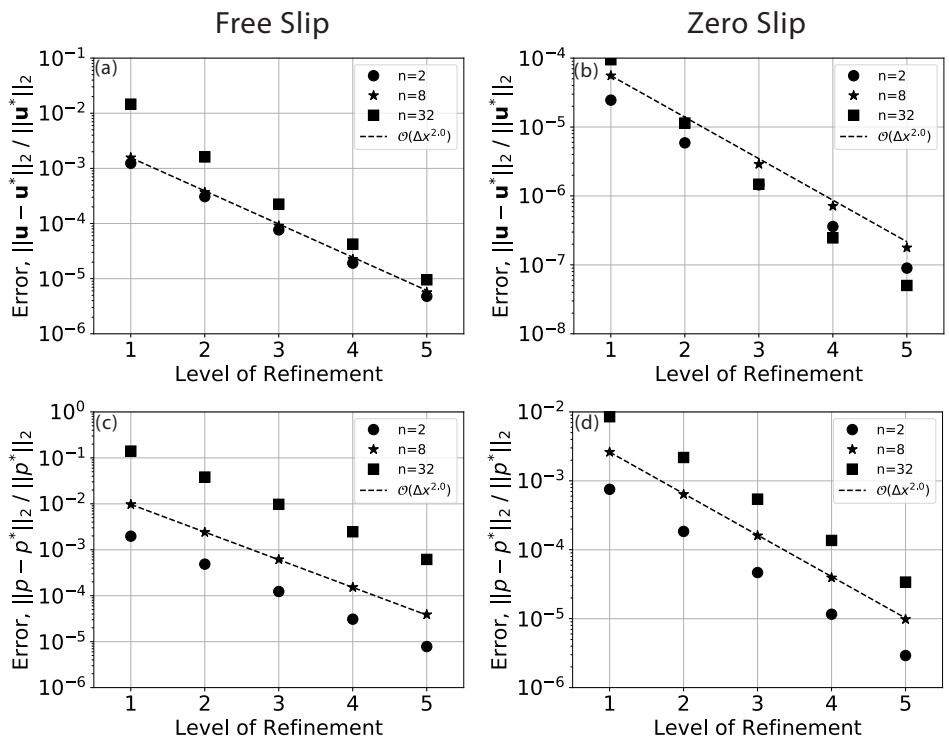

**Figure 5.** Convergence of velocity (a, b) and pressure (c, d), respectively, for smooth ($k = 2$) 2-D cylindrical cases with free-slip and zero-slip boundary conditions. Note that these cases do not incorporate an isoparametric approximation of the domain, hence the reduced convergence relative to comparable cases in Fig. 3.

giving a 3-D mesh consisting of 61440 tetrahedra. Again, resolution is doubled in all directions for subsequent refinement levels. In all cases non-dimensionalised coordinates were used with $R_- = 1.22$ and $R_+ = 2.22$, and the delta-function cases used $r' = (R_- + R_+)/2$. This choice of $r'$ ensures the density anomaly coincides with a grid layer at all mesh resolutions considered herein.

   In all figures, errors are given as relative errors, comparing the numerical solution, $\boldsymbol{u}$ and $p$, with the analytical solutions, $\boldsymbol{u}^*$
and $p^*$ (interpolated into the P2 and P1 function spaces respectively) in the $L_2$-norm

$$\frac{\|\boldsymbol{u} - \boldsymbol{u}^*\|_2}{\|\boldsymbol{u}^*\|_2} = \frac{\sqrt{\int_{\Omega_h} |\boldsymbol{u} - \boldsymbol{u}^*|^2}}{\sqrt{\int_{\Omega_h} |\boldsymbol{u}^*|^2}}, \quad \frac{\|p - p^*\|_2}{\|p^*\|_2} = \frac{\sqrt{\int_{\Omega_h} (p - p^*)^2}}{\sqrt{\int_{\Omega_h} (p^*)^2}}. \tag{65}$$

   Convergence plots for the 2-D cylindrical cases are presented in Fig. 3. With a smooth density profile we see optimal convergence for the P2-P1 element pair at third and second order for velocity and pressure respectively, with both free-slip and zero-slip boundary conditions. Cases with lower wave-number $n$ show smaller relative error than those at higher $n$, as expected.
The same observation holds for lower and higher polynomial order, $k = 2$ and $k = 8$, for the radial density profile. For the

free-slip and zero-slip delta-function cases however, convergence drops to 1.5 and 0.5 for velocity and pressure respectively. Furthermore, cases with lower $n$ do not consistently show smaller relative error than those at higher $n$.

We see similar results for the spherical results illustrated in Fig. 4: third and second order for velocity and pressure for the cases with a smooth density profile, with smaller relative errors for lower wave numbers $l$ and $m$. Note that here, the smooth vertical profile for density uses $k = l + 1$ in all cases. Again, for cases with a delta-function density anomaly, we observe a reduced order of convergence of 1.5 and 0.5 for velocity and pressure, respectively.

To examine the importance of an isoparametric approximation of the domain by a quadratic mesh, we ran the same cases with a linear mesh. The results are shown in Fig. 5 which demonstrates that the order of convergence of velocity in the smooth cylindrical cases is indeed limited to second order. The convergence of pressure remains at second order.

## 5 Discussion

### 5.1 Existing analytical benchmarks in shell domains

As indicated in the introduction, spherical delta-function cases, like those presented herein, have previously been used to validate global mantle convection codes (e.g. Zhong et al., 2008; Burstedde et al., 2013; Davies et al., 2013; Liu and King, 2019). These will be discussed in more detail in the following section.

The derivation for all 3-D cases in this paper rely on the Mie representation that decomposes the velocity solution into poloidal and toroidal components, through which, under the assumption of purely poloidal flow, the Stokes equations can be reduced to a biharmonic equation (10). Any solution to the inhomogeneous solution can then be combined with four linearly independent homogeneous solutions to this equation, the coefficients of which can be derived through the imposition of boundary conditions. For the smooth case with a generic monomial forcing term, the same decomposition (i.e. (45) and (44)), was used in Tosi and Martinec (2007) to derive the analytical solution for Stokes flow in two eccentrically nested spheres.

In Horbach et al. (2020) similar techniques were employed to derive benchmarks in spherical-shell domains that satisfy zero-slip and free-slip conditions and, in addition, a mixed zero-slip/free-slip case. Here the derivation starts by simply selecting four, in principle arbitrary, linearly independent solutions for the radially-dependent part of the poloidal scalar function. Again the imposition of boundary conditions fixes the coefficients of this linear combination. The corresponding right-hand side forcing term is then obtained by substitution.

The number of published benchmarks for 2-D cylindrical shell domains is more limited (e.g. Buffa et al., 2011; Blinova et al., 2016; Hoang et al., 2017). The derivation of the equivalent cases in 2-D cylindrical shell domains is somewhat simpler, but also ultimately relies on combining four independent homogeneous solutions and one inhomogeneous solution to a biharmonic equation.

In Blinova et al. (2016) analytical solutions in both cylindrical and spherical domains are presented for the Stokes equations with a radially-dependent viscosity. Because these are solutions in cylindrical or spherical coordinates without reference to any specific domain, they do not satisfy no-normal flow conditions on the boundary of a shell domain. They can be used in such domains as a numerical benchmark by specifying all velocity components of the analytical solution as a Dirichlet condition

for the model. Analytical solutions for radially-dependent viscosity were also presented in Thieulot (2017). Their solutions (in 3-D only) do satisfy no-normal flow conditions in a spherical-shell domain, but the tangential components are non-zero at the boundary, and thus still require inhomogeneous Dirichlet conditions. Spatially varying viscosity is of course an import aspect of mantle convection models for which these are effective benchmarks. The isoviscous solutions presented here, and those in Horbach et al. (2020), however, allow for the testing of zero-slip and free-slip conditions, where in particular free-slip conditions may pose various numerical challenges such as rotational modes, and, depending on the discretisation used, the (non-)alignment of velocity components with normal and tangential directions at the boundary.

## 5.2 Reduced order of convergence with discontinuous pressures

At first sight, the reduced order of convergence for the delta-function cases seems at odds with those expected for the P2-P1 element pair. However the mathematical proofs for the ideal order of convergence to solutions of the Stokes equations rely on certain regularity assumptions of the right-hand side forcing term and, related to that, on the regularity of the velocity and pressure solutions. The regularity of the delta function can be classified as being a member of the Sobolev space $H^{-1}(\Omega)$ the dual of the Sobolev space $H^1(\Omega)$, where for the sake of simplicity we assume $\Omega = \Omega_h$ in this section. This means that the delta function can be thought of as a continuous function

$$\delta_{r'} : v \mapsto \int_\Omega \delta(r - r') v(r, \phi) r \, dr \, d\phi = \int_\Omega v(r') r' \, d\phi, \tag{66}$$

which maps functions $v \in H^1(\Omega)$, the space of square integrable functions with square integrable weak derivatives, to $\mathbb{R}$. Girault and Raviart (2012) demonstrate that even with the very loose regularity condition that the right-hand side $f$ is in $H^m(\Omega)$ with $m \geq -1$, the Stokes equations in the weak form have a unique solution (given sufficient integral constraints) with velocity in $H^{m+2}(\Omega)$ and pressure in $H^{m+1}(\Omega)$. The analytical solutions derived here for the delta-case with $m = -1$, indeed have a discontinuous pressure in $H^0(\Omega) = L_2(\Omega)$, and a velocity with discontinuous normal derivative in $H^1(\Omega)$.

For velocity-pressure finite element pairs that satisfy the standard inf-sup, or LBB condition

$$\sup_{\boldsymbol{v} \in V} \frac{\int_\Omega \boldsymbol{v} \cdot \nabla q}{\|\boldsymbol{v}\|_1} \geq \beta \|q\|_2 \text{ for all } q \in W, \tag{67}$$

where $V$ and $W$ are the discrete vector and scalar function spaces respectively, and $\beta$ is a constant, it can be shown that the method converges and in fact

$$|\boldsymbol{u}^* - \boldsymbol{u}|_1 + \|p^* - p\|_2 \leq C_1 \left\{ \inf_{\boldsymbol{v} \in V} |\boldsymbol{u}^* - \boldsymbol{v}|_1 + \inf_{q \in W} \|p^* - q\|_2 \right\}, \tag{68}$$

where $C_1$ is a constant independent of $h$, $\boldsymbol{u}^*$ and $p^*$ are the exact solutions, $\boldsymbol{u}$ and $p$ the numerical solutions in the discrete function spaces $V$ and $W$ based on a mesh with mesh distance $h$, and $|\cdot|_1$ is the semi-norm

$$|\boldsymbol{u}|_1 = \sqrt{\sum_{i=1}^{\dim} \int_\Omega \|\partial_i \boldsymbol{u}\|^2}. \tag{69}$$

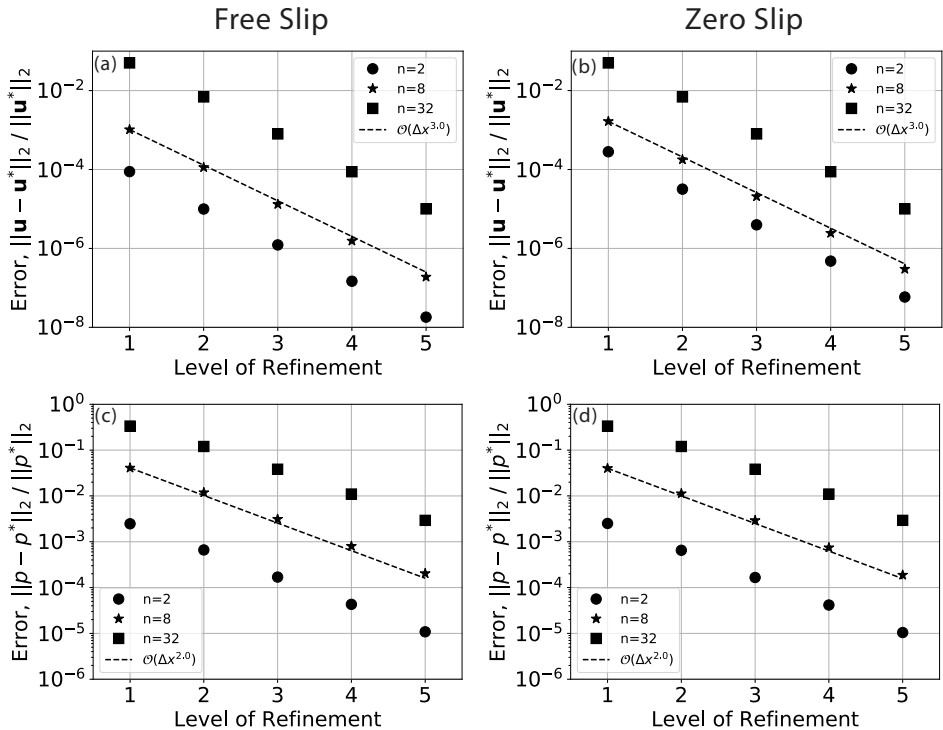

**Figure 6.** Convergence of velocity (a, b) and pressure (c, d), respectively, for delta-function 2-D cylindrical cases with free-slip and zero-slip boundary conditions with the P2$_{\text{bubble}}$-P1$_{\text{DG}}$ element pair.

In other words, the convergence of $\boldsymbol{u}$ and $p$ to $\boldsymbol{u}^*$ and $p^*$ is bounded by the best possible approximation of $\boldsymbol{u}^*$ and $p^*$ in the discrete spaces $V$ and $W$. For bounded functions with a discontinuity along a smooth interface, such as our analytical pressure solution $p^*$, the best approximation by continuous, piecewise-linear polynomials, i.e. $W = \text{P1}$, is bounded by

$$\inf_{q \in W} \|p^* - q\|_2 \leq C_2 h^{\frac{1}{2}} \|p^*\|_2, \tag{70}$$

(this bound can be derived from the order of convergence results in Bernardi, 1989). This therefore limits the convergence of the method.

A solution to this problem is found by allowing for discontinuities in the discrete pressure space. We demonstrate this here by considering $W = \text{P1}_{\text{DG}}$ the space of piecewise linear but discontinuous functions. To satisfy the inf-sup condition this

requires enriching the quadratic function space $P_2$ for velocity with a cubic bubble (Ern and Guermond, 2004; Boffi et al., 2013). Convergence results for this element pair are shown in Fig. 6. For the 2-D cylindrical delta-function cases, we observe the expected orders of convergence: third order for velocity and second order for pressure.

Finally, we compare our results with those presented in Zhong et al. (2008), Davies et al. (2013), Burstedde et al. (2013) and Liu and King (2019), who ran the same spherical cases with a delta function forcing, and found second order convergence

for velocity and second order convergence for pressure related diagnostics. It should be noted, however, that these studies only

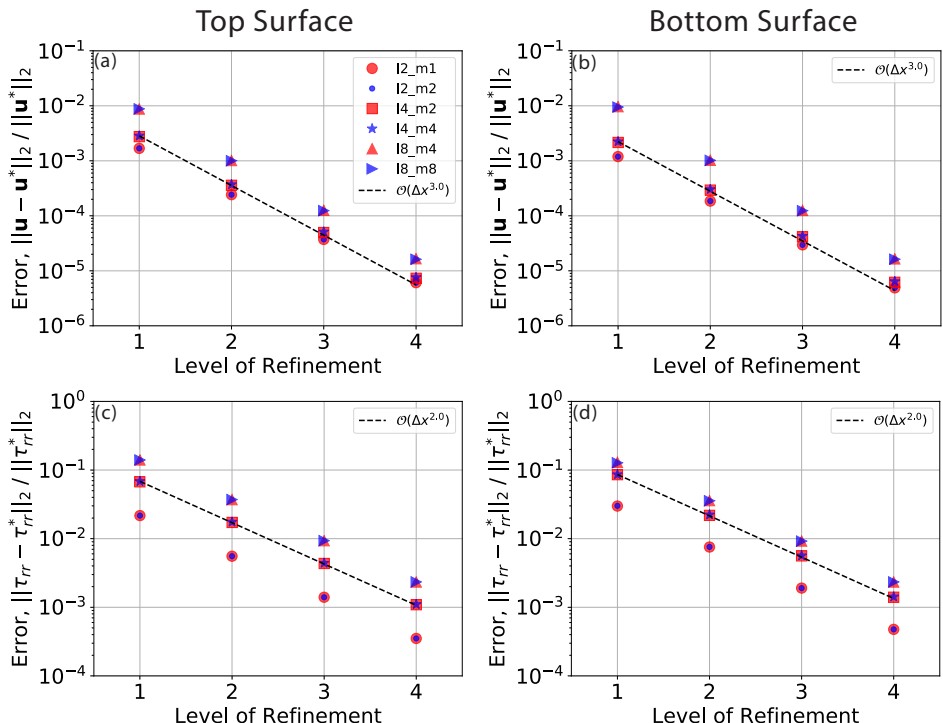

**Figure 7.** Convergence of velocity and radial stresses at top (a, c) top and (b, d) bottom surfaces ($r = R_\pm$), respectively, for delta-function 3-D spherical cases with free-slip boundary conditions.

examined surface diagnostics, such as the velocity divergence and the normal stress. When we examine comparable diagnostics, specifically, the relative error in velocity at the boundary and the boundary normal stress (illustrated in Fig. 7), we find velocity convergence at third order and normal stress at second order. It therefore appears that the reduced order of accuracy in the interior of the domain does not affect the surface response which still converges at the same order as for the smooth case.

The results of Zhong et al. (2008) (using CitcomS) and Davies et al. (2013) (using TERRA) were based on a Q1-P0 discretisation with a continuous piecewise trilinear velocity and piecewise constant, discontinuous pressure. Although our analysis above limits the convergence of $\|p - p^*\|_2$ for a P0 pressure $p$ to first order, second order super-convergence can be obtained in some cases by evaluating the analytical solution only in the cell centre. In other words, by comparing to a filtered piecewise constant analytical approximation $\bar{p}^*$, second order convergence can sometimes be observed in $\|p - \bar{p}^*\|_2$. For continuous pres-
sure approximations, such as the P2-P1 results in this paper, the Q1-Q1 discretisation of the Rhea model in Burstedde et al. (2013), and the Q2-Q1 discretisation of ASPECT in Liu and King (2019), reduced convergence in the interior of the domain is to be expected.

## 6    Conclusions

We have presented a series of 2-D cylindrical and 3-D spherical analytical solutions for the purpose of verifying mantle dynamics codes. These solutions are based upon either a delta-function density perturbation or a smooth forcing term, and we provide solutions for both free-slip and zero-slip boundary conditions. The combinations of dimension, forcing and boundary conditions, provide a series of eight analytical solutions that can be used as a basis for validating existing and future numerical codes, in cylindrical and spherical geometries. To facilitate this, we provide solutions in the form of a python package, Assess (*Analytical Solutions for the Stokes Equations in Spherical Shells*; Kramer, 2020).

We verify the convergence of the P2-P1 (Taylor Hood) finite element discretisation using Fluidity (Davies et al., 2011; Kramer et al., 2012). The continuous approximation of pressure can lead to a reduced order of convergence in the presence of discontinuities, which can be overcome using a discontinuous numerical approximation of pressure. It is important to note that this reduced order of convergence was only observed by comparing the numerical solution with the entire analytical solution in the interior of the domain. A comparison based on surface response only failed to highlight this issue.

*Code availability.*  The python package *Assess*, that implements the analytical solutions and evaluates them at arbitrary locations in the domain, is available from http://github.com/stephankramer/assess (see https://assess.readthedocs.io for documentation). An archived version is available from https://doi.org/10.5281/zenodo.3891545. To ensure correctness, both in the manuscript as in the python package, the coefficients for the various solutions are extracted from the LATEX-source automatically using SymPy (Meurer et al., 2017), verified to adhere to the equations using SageMath (The Sage Developers, 2019), and substituted in the python package. *Assess* has also been used to compute the errors in the Fluidity results in this paper. The Fluidity model, including source code and documentation, is available from https://fluidityproject.github.io/. All cases in this paper have been run with tag version 4.1.17, which is archived at https://doi.org/10.5281/zenodo.3988620.

*Author contributions.*  SCK derived the analytical solutions presented herein and updated Fluidity to run these cases. DRD ran the cases and prepared the corresponding figures. SCK wrote the manuscript with input from DRD and CRW. CRW implemented, set up and ran the P2$_{\text{bubble}}$-P1$_{\text{DG}}$ cases. All authors had significant input on the design and development of this research.

*Competing interests.*  The authors declare that they have no conflict of interest.

*Acknowledgements.*  DRD and SCK acknowledge support from the Australian Research Council, under grant numbers FT140101262 and DP170100058. SCK also acknowledges support from the UK Engineering and Physical Sciences Research Council (EPSRC) under grant number EP/R029423/1. Numerical simulations were undertaken on the NCI National Facility in Canberra, Australia, which is supported by

the Australian Commonwealth Government. We would also to thank Marcus Mohr for his very thorough review of the paper, and like to thank him and a second anonymous reviewer for their constructive feedback.

## Appendix A: Equations in polar coordinates

In this appendix we work out the incompressible Stokes equations in polar coordinates in terms of a streamfunction $\psi$, where the components of velocity are given by

$$u_r = -\frac{1}{r}\frac{\partial \psi}{\partial \varphi}, \quad u_\varphi = \frac{\partial \psi}{\partial r}, \tag{A1}$$

We make use of the following expressions for the derivatives of the unit vectors $\hat{r}$ and $\hat{\varphi}$ with respect to $r$ and $\varphi$

$$\hat{r}\cdot\nabla\hat{r} = \frac{\partial \hat{r}}{\partial r} = 0, \qquad\qquad \hat{\varphi}\cdot\nabla\hat{r} = \frac{1}{r}\frac{\partial \hat{r}}{\partial \varphi} = \frac{1}{r}\hat{\varphi}, \tag{A2}$$

$$\hat{r}\cdot\nabla\hat{\varphi} = \frac{\partial \hat{\varphi}}{\partial r} = 0, \qquad\qquad \hat{\varphi}\cdot\nabla\hat{\varphi} = \frac{1}{r}\frac{\partial \hat{\varphi}}{\partial \varphi} = -\frac{1}{r}\hat{r}. \tag{A3}$$

Using these we can work out the different components of stress

$$\tau_{rr} = 2\nu\hat{r}\cdot[\nabla u]\cdot\hat{r} = 2\nu\hat{r}\cdot\nabla(\hat{r}\cdot u) - 2\nu(\hat{r}\cdot\nabla\hat{r})\cdot u = 2\nu\frac{\partial u_r}{\partial r} - 0 = -2\nu\frac{\partial}{\partial r}\left(\frac{1}{r}\frac{\partial \psi}{\partial \varphi}\right), \tag{A4}$$

$$\begin{aligned}
\tau_{\varphi\varphi} = 2\nu\hat{\varphi}\cdot[\nabla u]\cdot\hat{\varphi} = 2\nu\hat{\varphi}\cdot\nabla(\hat{\varphi}\cdot u) - 2\nu(\hat{\varphi}\cdot\nabla\hat{\varphi})u &= \frac{2\nu}{r}\frac{\partial u_\varphi}{\partial \varphi} + \frac{2\nu}{r}\hat{r}\cdot u \\
&= \frac{2\nu}{r}\frac{\partial^2\psi}{\partial r\partial\varphi} - \frac{2\nu}{r^2}\frac{\partial\psi}{\partial\varphi} = 2\nu\frac{\partial}{\partial r}\left(\frac{1}{r}\frac{\partial\psi}{\partial\varphi}\right),
\end{aligned} \tag{A5}$$

$$\begin{aligned}
\tau_{r\varphi} = \nu\hat{r}\cdot[\nabla u]\cdot\hat{\varphi} + \nu\hat{\varphi}\cdot[\nabla u]\cdot\hat{r} &= \nu\hat{r}\cdot\nabla(u\cdot\hat{\varphi}) - \nu(\hat{r}\cdot\nabla\hat{\varphi})\cdot u + \nu\hat{\varphi}\cdot\nabla(u\cdot\hat{r}) - \nu(\hat{\varphi}\cdot\nabla\hat{r})\cdot u \\
&= \nu\frac{\partial}{\partial r}\left(\frac{\partial\psi}{\partial r}\right) - 0 + \frac{\nu}{r}\frac{\partial u_r}{\partial\varphi} - \frac{\nu}{r}\hat{\varphi}\cdot u \\
&= \nu\left(\frac{\partial^2\psi}{\partial r^2} - \frac{1}{r}\frac{\partial}{\partial\varphi}\left(\frac{1}{r}\frac{\partial\psi}{\partial\varphi}\right) - \frac{1}{r}\frac{\partial\psi}{\partial r}\right).
\end{aligned} \tag{A6}$$

Note that as expected $\tau_{rr} + \tau_{\varphi\varphi} = 0$. In the same way, we derive the following expression for the vorticity, or curl of the velocity

$$\text{curl }u = \hat{r}\cdot[\nabla u]\cdot\hat{\varphi} - \hat{\varphi}\cdot[\nabla u]\cdot\hat{r} = \frac{\partial^2\psi}{\partial r^2} + \frac{1}{r}\frac{\partial}{\partial\varphi}\left(\frac{1}{r}\frac{\partial\psi}{\partial\varphi}\right) + \frac{1}{r}\frac{\partial\psi}{\partial r} = \nabla^2\psi. \tag{A7}$$

The viscosity term in the Stokes equations can be written as

$$\nabla\cdot\tau = \nabla\cdot(\hat{r}\tau_{rr}\hat{r} + \hat{r}\tau_{r\varphi}\hat{\varphi} + \hat{\varphi}\tau_{\varphi r}\hat{r} + \hat{\varphi}\tau_{\varphi\varphi}\hat{\varphi}) \tag{A8}$$

$$= (\nabla\cdot\hat{r})(\tau_{rr}\hat{r} + \tau_{r\varphi}\hat{\varphi}) + \hat{r}\cdot\nabla(\tau_{rr}\hat{r} + \tau_{r\varphi}\hat{\varphi}) + (\nabla\cdot\hat{\varphi})(\tau_{\varphi r}\hat{r} + \tau_{\varphi\varphi}\hat{\varphi}) + \hat{\varphi}\cdot\nabla(\tau_{\varphi r}\hat{r} + \tau_{\varphi\varphi}\hat{\varphi}). \tag{A9}$$

In addition to (A3), we use the following identities

$$\nabla\cdot\hat{r} = \frac{1}{r}, \quad \nabla\cdot\hat{\varphi} = 0, \tag{A10}$$

and the fact that $\tau_{\varphi\varphi} = -\tau_{rr}$. After reordering to group the radial and transverse components this leads to

$$\nabla \cdot \tau = \left[ \frac{2}{r}\tau_{rr} + \frac{\partial \tau_{rr}}{\partial r} + \frac{1}{r}\frac{\partial}{\partial \varphi}\tau_{\varphi r} \right]\hat{r} + \left[ \frac{2}{r}\tau_{r\varphi} + \frac{\partial \tau_{r\varphi}}{\partial r} - \frac{1}{r}\frac{\partial \tau_{rr}}{\partial \varphi} \right]\hat{\varphi} \tag{A11}$$

$$= \left[ -\frac{2\nu}{r}\frac{\partial}{\partial \varphi}\frac{\partial^2 \psi}{\partial r^2} + \frac{1}{r}\frac{\partial}{\partial \varphi}\tau_{\varphi r} \right]\hat{r} + \left[ \nu\frac{\partial^3 \psi}{\partial r^3} + \frac{\nu}{r}\frac{\partial^2 \psi}{\partial r^2} - \frac{\nu}{r^2}\frac{\partial \psi}{\partial r} + \frac{\nu}{r^2}\frac{\partial^3 \psi}{\partial \varphi^2 \partial r} - \frac{2\nu}{r^3}\frac{\partial^2 \psi}{\partial \varphi^2} \right]\hat{\varphi} \tag{A12}$$

$$= -\frac{\nu}{r}\left[ \frac{\partial}{\partial \varphi}\nabla^2\psi \right]\hat{r} + \nu\left[ \frac{\partial}{\partial r}\nabla^2\psi \right]\hat{\varphi}. \tag{A13}$$

In combination with the pressure gradient term in polar coordinates

$$\nabla p = \frac{\partial p}{\partial r}\hat{r} + \frac{1}{r}\frac{\partial p}{\partial \varphi}\hat{\varphi}, \tag{A14}$$

we obtain the radial and transverse components of the Stokes momentum equation in (7) and (8).

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
