# Peer review of "Analytical solutions for mantle flow in cylindrical and spherical shells"

_Geoscientific Model Development, 2020_

## Short Comment (SC1) · 28 Sep 2020

Cedric THIEULOT

c.thieulot@uu.nl

Dear authors, In 2017 I carried out a similar exercise (Thieulot, Analytical solution for viscous incompressible Stokes flow in a spherical shell, Solid Earth, 8, 1181–1191, 2017 https://doi.org/10.5194/se-8-1181-2017). Although it looks like your (family of) solutions is more generic than mine, my solution allowed for a radial dependence of the viscosity. I was then wondering whether my solutions (in the case where viscosity is constant) are a subset of yours? Best regards, and congratulations for this impressive work and for sharing the code to compute the solution. Cedric.

---

## Author Comment (AC1) · 14 Oct 2020

Hi Cedric

Thanks for your very useful comment. Your paper (Thieulot, Analytical solution forviscous incompressible Stokes flow in a spherical shell, Solid Earth, 8, 1181–1191,2017 https://doi.org/10.5194/se-8-1181-2017) certainly looks relevant for our study and we should add a reference and briefly describe how it relates on revision of our paper.

Initially I also thought your isoviscous case should overlap with some of our solutions. In a sense I guess your solution starts from a different starting point by making a number of assumptions (section 2.1), to then, assuming the isoviscous case with m=-1, arrive at the density in eqn (50). In our paper on the other hand, we start from

assuming a generic form of density, e.g. eqn. (43) in our paper, to derive the solution from. This indeed suggests that your eqn. (50) could be seen as a linear combination of our "generic" solutions - at least for the last 2 terms: the first term $\sim \ln r/r^4$, is not a form we provide a solution for, but could probably be derived in a similar way. In terms of spherical harmonics, your cos(theta) corresponds to a degree l=1, and your assumption of azimuthal symmetry implies order m=1, i.o.w. Y_lm(theta,phi) = cos(theta) for l=1, m=0.

However then I realized is that our solution also makes different assumptions about boundary conditions. We provide solutions for both the free slip (i.e. zero tangential stress, and zero normal velocity), and no slip (all velocity components zero). If I understand correctly, in your solution the tangential stress follows from section 4.6, where g=0 at the boundary, so both tangential components would be (rf'-f)sin(theta), which is nonzero at the boundary. So I think our solutions in fact do not overlap, and describe different cases for different boundary conditions. In fact, obtaining solutions for "natural" boundary conditions, in particular free-slip which comes with a number of implementation issues such as strongly imposing a normal Dirichlet that is not aligned with the Cartesian direction, and associated rotational nullspace issues, was a strong motivation for our paper.

Of course your paper provides a wider family of solutions than just the isoviscous case, by offering solutions for radially dependent viscosity. As a future extension I think it would be interesting to see whether it is possible to derive solutions for the generic polynomial viscosity term you assume using the techniques in our paper

Best wishes Stephan Kramer

─────────────────────

---

## Referee Comment (RC1) · Marcus Mohr (Referee) · 28 Oct 2020

General Comments

The paper at hand derives families of analytic solutions for the incompressible Stokes problem with constant viscosity for 2D (annulus) and 3D (thick spherical shell). In each case a family of solutions is constructed for the case of a forcing term arising from a smooth density perturbation and an infinitely thin (delta-function) perturbation.

Based on these solutions numerical simulations are performed using different stable Finite Element pairs using the code Fluidity. The findings of these tests are discussed, providing especially a detailed analysis of convergence behaviour for the delta-function

case by means of Finite Element theory.

The reviewer wants to start by stating the he definitely enjoyed reading the paper. The mathematical derivation is sound and interesting. The numerical tests demonstrate the usefulness of the derived solutions for verfication of simulation software for global convection models, i.e. the choice of valid algorithms for discretisation and numerical solution of resulting linear systems as well as their correct implementation.

While the chosen model (incompressible, constant viscosity) is, naturally, not the most interesting one from the geodynamics perspective, solutions for this case are still valuable, as they allow to verify correctness for this baseline scenario.

The reviewer also especially liked the enlightening discussion of the convergence in the delta-function setting as this is a commonly used test scenario in publications in the field.

The only point of criticism is the following. While a large number of references are given for classical benchmarking studies that compare different codes against each other, there are only two citations, Zhong (1993), Kramer (2012) given involving analytical solutions. However, in recent years there have been several attempts to derive (semi)-analytic solutions for the simple and more intricate settings in both cartesian as well as cylindrical and spherical coordinates, see e.g. [BMP16,HMB20,PLP+14,Thi17], while a manufactured solution in spherical coordinates (for a ridge-like model) is also given in Burstedde (2013), cited in the paper. Thus, a brief explanation how the authors' contribution fits into these various approaches would be appreciated.

Specific Comments

- Equation (65) for computing the relative error: You are working with two different domains, $\Omega$ being the physical problem domain, and $\Omega_h$ the computational domain for the finite element method. I assume that the computation of the error

happens w.r.t. $\Omega_h$. In the isoparametric approach that does not make a significant difference, but for the affine mapping approach, Figure 5, I guess it would be better to make that distinction explicit.

- You treat the case of both boundaries being either zero-slip or free-slip. However, aren't many simulations run as a 'mixed' case, i.e. with Dirichlet boundary conditions for velocity (from plate reconstructions) on top and free-slip conditions at the CMB? In that sense, would it make sense to add (maybe in an appendix) also the coefficients for such a mixed case?

- p. 6, line 9 and equation (21): Shouldn't this read $R_- < r' < R+$ instead of $R_- \leq r' \leq R+$ and the piecewise $\psi$ be formulated for the two parts $R_- < r'$ and $r' < R_+$ only? Just to be mathematically more precise?

- Equations (25) and (26): To me the transition from (25) to (26) seemed mathematically quite involved, e.g. if one was to evaluate the integral in the right-hand side of (25) one would get a zero. Maybe you could add some additional details on this transition?

- p. 7, line 16: You are making use of the Mie representation of the velocity field. The necessary condition for this to have the form (27) is that the velocity field is *solenoidal*. For $R^3$ this is equivalent to $u$ being divergence-free. However, in the case of the target domain, the thick spherical shell, the two properties are not the same. You get that $u$ is solenoidal from $u$ being divergence-free and the fact the you have no outflow in the boundary conditions you consider. You might want to reformulate the sentence in this respect.

- If I have not missed it, you do not explicitly specify how $r'$ was chosen in you numerical tests. I assume that this is an interface between layers of the mesh already for the coarses mesh resolution used, isn't it? Could you please add that detail from completeness.

[Figure]

Out of curiosity, would you expect to observe oscillations in the convergence behaviour, if $r'$ was not a layer boundary on the meshes? I remember that in dipole modelling (geoelectricity and EEG simulation) people sometimes resort to special discretisation approaches for the $\delta$-function, e.g. St. Venant's principle, to avoid such issues.

- Sec. 3.4: The problem matrix representing the discretised Stokes system is singular independent of the type of boundary conditions due to the pressure only being determined up to an additive constant, isn't it? The free-slip conditions enhance the kernel of the matrix significantly leading to higher numerical effort (performing step (64) in each iteration seems required, while pressure needs only be adapted following (62) once in the end); you might want to make that clearer in line 15.

Suggestions

- p. 4, line 19 and p. 9, line 9: A reference for the biharmonic equation resp. its solutions might be helpful for the general audience; it is not quite as common knowledge as the harmonic equation and its solutions

- p. 4, line 14: You might consider changing 'top and bottom' to 'inner and outer' for the cylindrical domain

- p. 6, line 18: Shouldn't the strip be defined as $(r' - \epsilon, r' + \epsilon) \times (0, 2\pi)$?

- p. 9, line 3: The standard definition of the term domain in calculus is *an open and connected set*, so *connected domain* sounds like a pleonasm ;-)

- p. 4, equation (19) and line 19: I must admit that as I reader I very much behave like a one-pass-compiler, as soon as I encounter something I cannot follow I stumble. For the sake of people like me you might consider moving that sentence

which explains why there's only a $\cos$ term in (19) up front. Also you might state that you consciously neglect other harmonic functions (such as $\ln(r)$), for similar reasons as given in line 13.

- p. 1, line 2: IMHO 'within' does not sound quite proper here?

- In your paper you are using the term 'natural' boundary condition. If I understand correctly, you mean 'inspired by nature'? I am asking as in classical FE analysis there is that distinction between 'natural' and 'essential' boudnary conditions, so I was at first glance a little confused. Maybe change it to 'physical', if that still fully expresses what you want to convey.

- p. 1, line 24: Aren't 3D global mantle convection models being simulated routinely today, and not only *becoming more common*? I mean, you site references from the last 35 years ;-)

- equation (A7): Is there any specific motivation for defining the 2D polar curl in this way? It seems to be just the negative of what one would obtain by constantly extending a 2D field in z-direction and taking the z-component of the curl in 3D cylindrical coordinates?

Questions out of curiosity

- I found your discussion of the reason for the reduced convergence rates in the $\delta$-function case with the $P_2 - P_1$ Taylor-Hood element very interesting. As (68) only contains the $H_1$ semi-norm of $u$, is there an easy way to see from (70) why we get a similar $\frac{2}{3}$ order reduction in the $H_0$ norm of velocity as we do for pressure? Maybe I am missing some standard FE-analysis argument?

- Do you have any (speculative) idea why that convergence issue is not observed when one only examines surface quantities?

- In your 3D test cases you always select two similar combinations of degree and order $(\ell, m)$, which is $m = \ell$ (sectoral) and $m = \ell/2$ (tesseral); I was a little surprised that the errors seem to be fully identical for the two choices, because the number and direction of nodal lines differs. Are the differences just too small to be visible in the figures? Can you comment on that?

Technical Corrections

- equation (14): please check sign, I might have miscalculated, but I think it should read $H_n = + \dots$
- p. 6, line 15: 'expect a continuity' → 'expect continuity'
- p. 21, line 12: solution → solution(s)
- equation (A3): $\hat{\varphi} \cdot \nabla \hat{\varphi} = \dfrac{1}{r} \dfrac{\partial \hat{r}}{\partial \varphi} \to \dfrac{1}{r} \dfrac{\partial \hat{\varphi}}{\partial \varphi}$
- equation (A12): spurious $+$ near end of equation
- reference Hernlund, Tackley, 2007: IMHO that should be 2008

**References**

Irina Blinova, Ilya Makeev, and Igor Popov. Benchmark Solutions for Stokes Flows in Cylindrical and Spherical Geometry. *Bull. Transilvania U. Brasov*, 9(1):11–16, 2016.

André Horbach, Marcus Mohr, and Hans-Peter Bunge. A Semi-Analytic Accuracy Benchmark for Stokes Flow in 3-D Spherical Mantle Convection Codes. *GEM - International Journal of Geomathematics*, 11(1), 2020.

I. Yu. Popov, I. S. Lobanov, S. I. Popov, A. I. Popov, and T. V. Gerya. Practical analytical solutions for benchmarking of 2-d and 3-d geodynamic stokes problems with variable viscosity. *Solid Earth*, 5:461–476, 2014.

[Figure]

Cedric Thieulot. Analytical solution for viscous incompressible Stokes flow in a spherical shell. *Solid Earth*, 8:1181–1191, 2017.

---

## Referee Comment (RC2) · Anonymous Referee #2 · 25 Jan 2021

Summary:

With the spirit of providing a set of benchmark functions for mantle flow simulation finite element codes, the authors compute explicitly a set of analytical solutions satisfying the incompressible Stokes equations, together with free-slip or zero-slip boundary conditions over both, 2D cylindrical and 3D spherical domains. The authors describe in detail the deduction of these analytical solutions in the several scenarios and validate them by performing a study of the FEM convergence rates considering the finite element framework Fluidity, obtaining the expected convergence rates when considering the well-known Taylor-Hood elements. This is complemented with a discussion on mathematical details of finite element convergence rates to give fundamentals of the suboptimality obtained when considering the delta distribution as source term. The

authors also provide a python package (Assess) allowing for evaluations of the analytical solutions at arbitrary points of the corresponding domain. Despite some minor typos, the document is in general well written and structured. Even if I did not verify in detail all the computations carried out by the authors to obtain the analytical solutions, their deduction seems correct and the numerical experimentation correctly verifies their accuracy. Moreover, the work perfectly fits with the scope of the journal since the authors also provide a python package with the implementation that can be used for the verification of any computational model based on the resolution of the incompressible Stokes equations.

Recommendation:

At a first glance, I am inclined to suggest a major review since I am not totally convinced of its novelty. In particular, it must be verified by the authors if this set of solutions have not already been considered in existing works. For instance, one of the articles mentioned by the other review [2] already includes a discussion of analytical solutions for both cases of boundary conditions considered by the authors, plus the mixed BC case. Additionally, [4] also considers the benchmarks for the incompressible Stokes equations in a 3D spherical shell. Once this is verified, including an extensive state-of-the-art literature review, I will be happy recommending it for publication in GMD.

Additional suggestions:

I have some additional suggestions listed below that, with the spirit of complementing the already ongoing discussion with the other review, Âă as it is already a very detailed review in my opinion.

1. The abstract and introduction can be improved by including self-explained sentences and letting citations only for verification purposes. In particular, the sentences "Computational models of mantle ..." in the abstract, and "3-D spherical geometry is implicitly required to simulate global mantle dynamics" in the introduction must be complemented with a brief explanation, from the physical and numerical point of view, of

the loosing when considering a cartesian model of the globe.

2. I strongly support the suggestion of considering a more deep literature review. In particular it is also missing a discussion of the already existing benchmarks in the FE community for Stokes equations in smooth domains (e.g., [1,3]). 3.- Please review the punctuation of the entire document. In particular, equations must be treated as part of the text. For instance, equations (6), (9), (10)(this is a typo), (12), ... must be ended with a "dot". 4.- In line 20, page 4 add "s" to relation

5.- In line 17, pag 5 (and in the rest of the article when it corresponds) add "coefficients" after "solution". i.e., write "solution coefficients"Âă

References:

[1] Buffa, A., de Falco, C., Sangalli, G.: IsoGeometric Analysis: Stable elements for the 2D Stokes equation. International Journal for Numerical Methods in Fluids 65(11-12), 1407–1422 (2011).

[2] Horbach, A., Mohr, M., Bunge, HP. A semi-analytic accuracy benchmark for Stokes flow in 3-D spherical mantle convection codes. Int J Geomath 11, 1 (2020).

[3] Hoang, T., Verhoosel, C.V., Auricchio, F., van Brummelen, E.H., Reali, A.: Mixed Isogeometric Finite Cell Methods for the Stokes problem. Computer Methods in Applied Mechanics and Engineering 316, 400 – 423 (2017).Âă

[4] Liu, S., King, S.D.: A benchmark study of incompressible Stokes flow in a 3-D spherical shell using ASPECT. Geophysical Journal International 217(1), 650–667 (2019).

---

## Author Comment (AC2) · 12 Feb 2021

Thank you for your very thorough, careful and constructive review of our study. We are pleased to know you enjoyed reading the paper. In response to your main point of criticism, we agree that the paper should include some more (recent) references and a discussion of other work containing analytical Stokes solutions. Most relevant here is HMB20, your very nice paper with Andre Horbach and Hans-Peter Bunge, which came out last year and we had not yet seen. The two papers (i.e. ours and yours) use similar techniques to derive non-trivial free slip and no-slip analytical Stokes solutions in spherical shells. In our paper, we derive the analytical Stokes solution for a simple (radially monomial) forcing combined with spherical harmonics. This is achieved by decomposing the solution using the Mie representation, and deriving a biharmonic equation for

poloidal part. An inhomogeneous solution is combined with four solutions to the homogeneous equation, the coefficients of which are determined by imposing boundary conditions. In contrast, HMB20 describes a procedure starting with the selection of four arbitrarily chosen, linearly independent, solutions for the poloidal function, after which a linear combination of these is determined through the boundary conditions. Finally the necessary right-hand-side term is then constructed by substitution of the corresponding velocity field into the Stokes equations. In a sense, this approach is more akin to the method of manufactured solutions (MMS), but overcomes its usual problem of not satisfying desired boundary conditions. Both approaches are fairly flexible in terms of selecting solutions with desired physical properties, e.g. selecting high order polynomial to obtain strong gradients near the top boundary. In addition, our paper provides similar solutions in cylindrical shell domains, and a different set of solutions, both in 2D and 3D, based on an infinitely thin density anomaly. For the latter, for which the spherical solutions have been used previously as benchmarks [Zhong'08 and others], we discuss a particular issue with the continuous Galerkin finite element method. We believe the two papers are therefore highly complementary.

In the revised version of the manuscript we have given more attention to this and other references you mention, by referring to them in the introduction and by adding a discussion at the end of the manuscript that compares ours with previously published analytical spherical Stokes solutions. In our response to Cedric Thieulot we already indicated that we believe the analytical solution in Thi17 (isoviscous case) does not satisfy no-slip or free slip boundary conditions and therefore does not overlap with ours. The same appears to be true for the solutions in BMP16. PLP+14, which indeed also deserves a mention, deals with Cartesian domains only.

**Specific Comments**

- *Equation (65) for computing the relative error: You are working with two different*

*domains, $\Omega$ being the physical problem domain, and $\Omega_h$ the computational domain for the finite element method. I assume that the computation of the error happens w.r.t. $\Omega_h$. In the isoparametric approach that does not make a significant difference, but for the affine mapping approach, Figure 5, I guess it would be better to make that distinction explicit.*

As highlighted, we did not make a proper distinction between the (idealised) physical domain and the numerical domain. This has been corrected by introducing the notation $\Omega$ and $\Omega_h$ respectively, as suggested.

- *You treat the case of both boundaries being either zero-slip or free-slip. However, aren't many simulations run as a 'mixed case', i.e. with Dirichlet boundary conditions for velocity (from plate reconstructions) on top and free-slip conditions at the CMB? In that sense, would it make sense to add (maybe in an appendix) also the coefficients for such a mixed case?*

We agree that solutions with mixed boundary conditions, free slip at the bottom and a prescribed, non-zero Dirichlet boundary condition for velocity at the top, inspired by kinematically-driven models (from plate motion histories), are very much of interest as well. The derivation of another such case would be fairly straightforward in principle, but there is a non-trivial amount of practical work involved. The procedure would be similar to the zero-slip and free-slip cases presented, albeit with some adjustments to the conditions imposed, where the solution can again be derived automatically using sympy or sage. What is far more involved is to transform the often unwieldy sympy/sage solutions into something that can be compactly expressed in a manuscript. The required hand-editing of the solutions is the reason why we introduced automated tests that extract the final latex expression of the solutions and checks their correctness. More importantly we think that adding just a single simple case, the most obvious would be a rigid rotation around a single axis, would be of limited value. To properly test the capability to simulate plate-driven models would require benchmarks that include shear, convergence and divergence in the prescribed velocity. We would be willing to consider, if the interest arises, to include such cases in the assess python package (where readability is of less concern), but we believe adding an appendix with the fully worked out solution would add little to the reader of the manuscript.

- *p. 6, line 9 and equation (21): Shouldn't this read $R_- < r' < R_+$ instead of $R_- \leq r' \leq R_+$ and the piecewise $\psi$ be formulated for the two parts $R_- < r'$ and $r' < R_+$ only? Just to be mathematically more precise?*

  This has been corrected as suggested.

- *Equations (25) and (26): To me the transition from (25) to (26) seemed mathematically quite involved, e.g. if one was to evaluate the integral in the right-hand side of (25) one would get a zero. Maybe you could add some additional details on this transition?*

  The transition between (25) and (26) was indeed not entirely mathematically sound. This was the consequence of a failed attempt to simplify the argument. We have now corrected the derivation of (26) and inserted two additional steps.

- *p. 7, line 16: You are making use of the Mie representation of the velocity field.The necessary condition for this to have the form (27) is that the velocity field is solenoidal. For $R^3$ this is equivalent to $u$ being divergence-free. However, in the case of the target domain, the thick spherical shell, the two properties are not the same. You get that $u$ is solenoidal from $u$ being divergence-free and the fact the you have no outflow in the boundary conditions you consider. You might want to reformulate the sentence in this respect.*

  We have clarified that the condition for Mie's representation is a solenoidal velocity which indeed follows from divergence freeness in combination with the no normal flow conditions in our domains.

- *If I have not missed it, you do not explicitly specify how $r'$ was chosen in you numerical tests. I assume that this is an interface between layers of the mesh already for the coarses mesh resolution used, isn't it? Could you please add that detail from completeness.*

  *Out of curiosity, would you expect to observe oscillations in the convergence behaviour, if $r'$ was not a layer boundary on the meshes? I remember that in dipole modelling (geoelectricity and EEG simulation) people sometimes resort to special discretisation approaches for the $\delta$-function, e.g. St. Venant's principle, to avoid such issues.*

  The value that was used for $r'$ (and in fact $R_-$ and $R_+$) was indeed missing in the paper. They were: $R_- = 1.22$, $r' = 1.72$, $R_+ = 2.22$ following a standard non-dimensionalisation in which the Mantle depth is one. In other words, $r'$ was always halfway between top and bottom, and indeed always coinciding with a grid level. This allows us to treat the contribution from the infinitely thin forcing as a surface integral as explained in section 3.3. Other authors [e.g. Zhong et al. '08] have approximated the delta function with a finite element basis function with a support over the two adjacent cells and a grid-dependent amplitude that goes to infinity as $h \to 0$. In local experiments I have seen similar convergence issues with this kind of forcing, and I suspect the same would be true if the location were not grid-aligned. It might be that other ways of smoothing the forcing would improve the convergence. Fundamentally though, convergence at the ideal order would not be guaranteed through the classical finite element analysis.

- *Sec. 3.4: The problem matrix representing the discretised Stokes system is singular independent of the type of boundary conditions due to the pressure only being determined up to an additive constant, isn't it? The free-slip conditions enhance the kernel of the matrix significantly leading to higher numerical effort (performing step (64) in each iteration seems required, while pressure needs only be adapted following (62) once in the end); you might want to make that clearer*

*in line 15.*

The combined velocity pressure saddle point system is indeed singular in all cases (see also line 6, page 14) The projections in equations (62) **and** (63) are big L2-projections and are both only performed once after the entire solving process has finished. To clarify lines 14-15 on page 14: 'It should be noted that the same modes...'; This sentence should really start a new paragraph and 'these modes' refer to both the pressure and velocity modes described just before that. The corresponding (little) l2-projections are performed in the iterative solution process performed by PETSc, which consists of an outer (FGMRES) iteration solving the Schur complement equation in which the required matrix vector multiplication with the inverse of the velocity block induces an inner (CG) iteration. The little l2 projection corresponding to the velocity modes are subtracted every inner iteration. The little l2 projection corresponding to constant pressure is subtracted every outer iteration. We hope this clears up the final one of your comments. We have tried to clarify this paragraph in revision. Without going into to much details on the solver strategy (which is also described in Davies et al. '11), the point we try to bring across is that the null-modes you have to provide to ensure convergence of the iterative solvers, which because of their abstraction are typically formulated in an l2-inner product context, are insufficient to ensure convergence to the analytical solution because of the difference between l2 and L2 projections which in particular for non-uniform meshes can be significant.

**Suggestions:**

- *p. 4, line 19 and p. 9, line 9: A reference for the biharmonic equation resp. its solutions might be helpful for the general audience; it is not quite as common knowledge as the harmonic equation and its solutions*

- *p. 4, line 14: You might consider changing 'top' and 'bottom' to 'inner and outer' for the cylindrical domain*

- *p. 6, line 18: Shouldn't the strip be defined as $(r' - \epsilon, r' + \epsilon) \times (0, 2\pi)$?*

- *p. 9, line 3: The standard definition of the term domain in calculus is an open and connected set, so connected domain sounds like a pleonasm ;-)*

- *p. 4, equation (19) and line 19: I must admit that as I reader I very much behave like a one-pass-compiler, as soon as I encounter something I cannot follow I stumble. For the sake of people like me you might consider moving that sentence which explains why there's only a $\cos$ term in (19) up front. Also you might state that you consciously neglect other harmonic functions (such as $\ln(r)$), for similar reasons as given in line 13.*

- *p. 1, line 2: IMHO 'within' does not sound quite proper here?*

- *In your paper you are using the term 'natural' boundary condition. If I understand correctly, you mean 'inspired' by nature? I am asking as in classical FE analysis there is that distinction between 'natural' and 'essential' boundary conditions, so I was at first glance a little confused. Maybe change it to 'physical', if that still fully expresses what you want to convey.*

- *p. 1, line 24: Aren't 3D global mantle convection models being simulated routinely today, and not only becoming more common? I mean, you site references from the last 35 years ;-)*

- *equation (A7): Is there any specific motivation for defining the 2D polar curl in this way? It seems to be just the negative of what one would obtain by constantly extending a 2D field in z-direction and taking the z-component of the curl in 3D cylindrical coordinates?*

Your list of suggestions are all very much to the point and have been incorporated in the revised manuscript. Regarding, the sign in the 2D curl definition in the appendix this is indeed not standard and not as we intended. What I think happened was that I worked back from the curl of the streamfunction with a different sign convention for the streamfunction in mind. In any case we have corrected this to follow the usual definition. This does not affect the derivations in the rest of the paper.

**Questions out of curiosity**

- *I found your discussion of the reason for the reduced convergence rates in the $\delta$-function case with the $P_2 - P_1$ Taylor-Hood element very interesting. As (68) only contains the $H_1$ semi-norm of $u$, is there an easy way to see from (70) why we get a similar $\frac{2}{3}$ order reduction in the $H_0$ norm of velocity as we do for pressure? Maybe I am missing some standard FE-analysis argument?*

  I believe the next step, to derive an L2 error bound for u out of the H1 seminorm bound, is the so called Aubin-Nitsche trick. The most compact write up that I could find is (4.1)-(4.3) in Verfuehrt'84. Here, in turn, you need to find a bound in the $H^1$-norm for the solution $z$ and numerical solution $z_h$ of an arbitrary right-hand side $v$ in $L_2$ (rather than $H^{-1}$) for which, with a domain of the right regularity, you can find a solution $v$ in $H^2$ (instead of $H^1$). The $H^1$-norm of the difference between $z$ and $z_h$ is then first order in $h$. This together with the $h^{1/2}$ bound we already had, gives you the expected $h^{3/2}$ (see also Theorem 1.9 in Girault). Note that this relies on having this $h^{1/2}$ bound in the first place which I believe isn't guaranteed for arbitrary $H^{-1}$ right-hand sides (hence Theorem 1.9 in Girault is only valid for H2 velocity solutions, i.e. the original rhs should be L2). The bound in (70) is based on the assumption of a single jump in pressure where the number of simplices affected by this jump scales with $1/h^{\dim-1}$ (i.e. occurs along a dim-1 "front" that can be well resolved).

- *Do you have any (speculative) idea why that convergence issue is not observed when one only examines surface quantities?*

I am speculating that a similar bound to (68) is possible where the numerical error in the solution restricted to the boundary is related to the best possible approximation on the boundary for the given discrete function space. Note that the bound in (70) for the best possible approximation of the analytical solution $p$ is actually just bounded by the (worst) regularity of $p$ in the triangles near the jump (you can derive it from the generic finite element approximation error bounds in Bernardi '89). However this does not mean that the numerical solution of the discretised PDE has its $h^{1/2}$ error concentrated in these cells; this I've tried: even if you restrict the $L_2$ error integral to integrate only over parts of the domain well away from the jump, the convergence is still $h^{1/2}$. So I suspect this really only works for an integral just over the boundary, or a volume integral at some limited distance from the boundary where that distance scales to zero as $h \to 0$.

- *In your 3D test cases you always select two similar combinations of degree and order $(l, m)$, which is $m = l$ (sectoral) and $m = l/2$ (tesseral); I was a little surprised that the errors seem to be fully identical for the two choices, because the number and direction of nodal lines differs. Are the differences just too small to be visible in the figures? Can you comment on that?*

We believe this is just an artefact of the choice of $k$, the degree of the radial dependence, where the combinations you mention that seems to have similar error, also have the same degree $k$. So it seems that for the ratio of tangential and radial resolution we have chosen the tangential variation is well resolved and the error is probably more restricted by the vertical resolution.

**Technical Corrections**

- *equation (14): please check sign, I might have miscalculated, but I think it should read $H_n = +\ldots$*

- *p. 6, line 15: 'expect a continuity' → 'expect continuity'*

- *p. 21, line 12: solution → solution(s)*

- *equation (A3): $\hat{\varphi} \cdot \nabla \hat{\varphi} = \frac{1}{r}\frac{\partial \hat{r}}{\partial \varphi} \quad \rightarrow \quad \frac{1}{r}\frac{\partial \hat{\varphi}}{\partial \varphi}$*

- *equation (A12): spurious $+$ near end of equation*

- *reference Hernlund, Tackley, 2007: IMHO that should be 2008*

I have double checked (14), and I must admit at first I came to the same conclusion of a $+$ in the $H_n$ equation. This confused me a lot as we had been using a minus sign in our convergence analysis. Luckily, after going through the derivation once more, and more carefully, I am now convinced the original minus sign is correct.

For completeness let me include 'my working'. Assume $p = H_n r^{-n} \cos(n\varphi)$ and $\psi = D_n r^{-n+2} \sin(n\varphi)$, then

$$
\begin{aligned}
\frac{\partial p}{\partial r} &= -n r^{-n-1} H_n \cos(n\varphi), \\
\frac{1}{r}\frac{\partial p}{\partial \varphi} &= -n r^{-n-1} H_n \sin(n\varphi), \\
\nabla^2 \psi &= \left((-n+2)^2 - n^2\right) D_n r^{-n} \sin(n\varphi) \\
&= -4(n-1) D_n r^{-n} \sin(n\varphi), \\
\frac{\nu}{r}\frac{\partial \nabla^2 \psi}{\partial \varphi} &= -4\nu n(n-1) D_n r^{-n-1} \cos(n\varphi),
\end{aligned}
$$

$$-\nu\frac{\partial\nabla^2\psi}{\partial r} \quad = -4\nu n(n-1)D_n r^{-n-1}\sin(n\varphi).$$

So that equations (7) and (8) become respectively:

$$-n\left[4\nu(n-1)D_n + H_n\right]r^{-n-1}\cos(n\varphi) \quad = 0,$$
$$-n\left[4\nu(n-1)D_n + H_n\right]r^{-n-1}\sin(n\varphi) \quad = 0$$

which indeed implies that $H_n = -4\nu(n-1)D_n$ as in equation (14).

We would like to thank the reviewer, once again, for their very thorough and constructive review of our derivations and the manuscript. It was clearly a huge effort and we are very grateful. The other technical corrections have been included in the revised manuscript.

**References**

References HMB20, Thi17, BMP16, PLP14 as in your referee comment.

Bernardi '89: Christine Bernardi, "Optimal Finite-Element Interpolation on Curved Domains", SIAM Journal on Numerical Analysis 26.5 (1989): 1212-1240, https://doi.org/10.1137/0726068

Verfuhrt '84: R. Verfuhrt, "A combined conjugate gradient - multi-grid algorithm for the numerical solution of the Stokes problem", IMA J Numer Anal, Volume 4, Issue 4, 1984, Pages 441–455, https://doi.org/10.1093/imanum/4.4.441

All other references appear in the submitted paper.

———————————————

---

## Author Comment (AC3) · 12 Feb 2021

We thank the reviewer for their comments. As we have indicated in our response to the first reviewer (Mohr), we agree that more references should have been included to reflect more recent studies with analytical Stokes solutions in shell domains. We have addressed this in a revised manuscript by adding references to these more recent publications and by adding a new section that discusses the relation between the solution sets in this paper and those publications. We thank the reviewer for highlighting this shortcoming of our original manuscript.

Regarding the novelty of the paper, the mathematical techniques that were used in the paper are classical and have been used elsewhere, e.g. to derive the solutions in

the propagator matrix method that is used in the Zhong '08 benchmark. The solution for that case corresponds to one of the eight cases that we present in the paper, and the solution coefficients for these have appeared elsewhere, e.g. Ribe '09, which we cite. The solutions for the other seven cases we have not seen derived and presented elsewhere. Although the solution coefficients are simply given after our summary of the classical theory to derive these, the actual derivation is laborious and error-prone. We therefore believe these to be of significant value for the geodynamic modelling community, in particular in combination with a software implementation in the form of the assess python package. We note that reviewer 1 and Cedric Thielot, both world leading geodynamical modellers, agree with this viewpoint.

In addition, although the techniques for deriving analytical Stokes solutions in spherical domains are more familiar in this community, the equivalent derivations in 2-D cylindrical shell domains are less well known. The mathematical techniques for deriving these are not new, but we believe there is value to the community in our comprehensive overview. Finally, we discuss an issue that has been overlooked in all recent global mantle convection validation papers, which is that models based on a continuous pressure discretisation have a very poor convergence for cases with a delta-function forcing in the interior of the domain, which will not be noticed if one only examines the surface response.

Of the additional references you provide, [2] appeared just before we submitted our paper. In our response to Marcus Mohr, one of the co-authors of the paper, and in the new discussion section at the end of our revised manuscript, we have indicated how our solutions relate to theirs. The ASPECT paper [4] uses the same Zhong '08 test case we already mentioned. We have now added this paper [4], in our discussion alongside three other papers that have used the same approach (CitcomS, Rhea and TERRA). References [1] and [3] indeed, contain cylindrical test cases for Stokes flow, but [3] only contains a Method of Manufactured Solutions (MMS) case, the limitations of which we have discussed in our paper; [1] appears to contain a test case with a

driven cavity-like solution in an annulus segment, however it does not actually present the analytical solution, other than in a figure, or how it has been derived. Nonetheless, for completeness we have added references [1,3] to our modified discussion section.

Additional suggestions:

1. *The abstract and introduction can be improved by including self-explained sentences and letting citations only for verification purposes. In particular, the sentences "Computational models of mantle ..." in the abstract, and "3-D spherical geometry is implicitly required to simulate global mantle dynamics" in the introduction must be complemented with a brief explanation, from the physical and numerical point of view, of the loosing when considering a cartesian model of the globe.*

   We have re-read the abstract and introduction and disagree with the reviewer here. The first sentence of the abstract is concise and to the point - with further information provided in the introduction. Furthermore, we do not think that a qualifier is necessary for the second suggestion. The mantle is a 3-D spherical shell: simulations of 'global' mantle dynamics therefore cannot be undertaken in Cartesian domains. We are also unclear what the reviewer means by "cartesian model of the globe". Perhaps the reviewer is referring to structured, "cubed-sphere" approaches? Regardless of the numerical approach, the physics of global mantle convection takes place in a 3-D spherical-shell geometry. We have rephrased the second sentence to make it more clear that we are referring here to the step between processes in a Cartesian physical domain, and global mantle convection models.

2. *I strongly support the suggestion of considering a more deep literature review. In particular it is also missing a discussion of the already existing benchmarks in the FE community for Stokes equations in smooth domains (e.g., [1,3]).*

Discussed above and addressed in our revised manuscript.

3. *Please review the punctuation of the entire document. In particular, equations must be treated as part of the text. For instance, equations (6), (9), (10)(this is a typo), (12), ... must be ended with a "dot".*

   We have reviewed and corrected the punctuation of all equations as suggested. Thank you for highlighting this issue in our original submission.

4. *In line 20, page 4 add "s" to relation*

   Corrected.

5. *In line 17, page 5 (and in the rest of the article when it corresponds) add "coefficients" after "solution". i.e., write "solution coefficients"*

   We have followed the reviewer's suggestion throughout.